# Non-asymptotic Convergence of Training Transformers for Next-token Prediction

**Ruiquan Huang**
Penn State University
State College, PA, 16801
rzh5514@psu.edu

**Yingbin Liang**
Ohio State University
Columbus, OH, 43210
liang.889@osu.edu

**Jing Yang**
Penn State Univeristy
State College, PA, 16801
yangjing@psu.edu

## Abstract

Transformers have achieved extraordinary success in modern machine learning due to their excellent ability to handle sequential data, especially in next-token prediction (NTP) tasks. However, the theoretical understanding of their performance in NTP is limited, with existing studies focusing mainly on asymptotic performance. This paper provides a fine-grained non-asymptotic analysis of the training dynamics of a one-layer transformer consisting of a self-attention module followed by a feed-forward layer. We first characterize the essential structural properties of training datasets for NTP using a mathematical framework based on partial orders. Then, we design a two-stage training algorithm, where the pre-processing stage for training the feed-forward layer and the main stage for training the attention layer exhibit fast convergence performance. Specifically, both layers converge *sub-linearly* to the direction of their corresponding max-margin solutions. We also show that the cross-entropy loss enjoys a *linear* convergence rate. Furthermore, we show that the trained transformer presents non-trivial prediction ability with dataset shift, which sheds light on the remarkable generalization performance of transformers. Our analysis technique involves the development of novel properties on the attention gradient and further in-depth analysis of how these properties contribute to the convergence of the training process. Our experiments further validate our theoretical findings.

## 1 Introduction

The transformer architecture (Vaswani et al., 2017) has revolutionized the field of machine learning, establishing itself as a foundation model for numerous applications, including natural language processing (NLP) (Devlin et al., 2018), computer vision (Dosovitskiy et al., 2020), and multi-modal signal processing (Tsai et al., 2019). In particular, transformers achieve tremendous empirical success in large language models (LLMs) such as GPT-3 (Brown et al., 2020). Despite the empirical success, limited theoretical understanding of transformers have caused a series of critical concerns about their robustness, interpretability, and bias issues (Bommasani et al., 2021; Belkin, 2024).

To overcome these issues, recent advances in transformer theory have investigated the convergence of training transformers under theoretically amenable setting such as linear regression (Mahankali et al., 2023; Zhang et al., 2023; Huang et al., 2023) and binary classification (Tarzanagh et al., 2023b,a; Vasudeva et al., 2024; Li et al., 2023). Nevertheless, one of the fundamental task in LLMs and other generative models is next-token prediction (NTP), which involves predicting the next word or token in a sequence, given the previous tokens. In NTP, a few recent theoretical studies have started to investigate the training dynamics of transformers (Tian et al., 2023a; Li et al., 2024). However, those works lack of fine-grained *non-asymptotic* convergence analysis of the training process, posing the following open questions for further investigation:

38th Conference on Neural Information Processing Systems (NeurIPS 2024).

*How fast does the training of a transformer converge in NTP?*

In addition, a pre-trained transformer empirically exhibits non-trivial generalization ability. A follow-up question from a theoretical point of view is that

*Can we show the generalization capability of a trained transformer on unseen data?*

In this paper, we take a first step towards addressing the aforementioned questions by studying the training dynamics of a single layer transformer consisting of a self-attention layer and a feed-forward layer for NTP. We summarize our contribution as follows.

- We develop a mathematical framework based on partial order to formally characterize the essential structural properties of the training dataset for next-token prediction. In particular, we introduce a realizable setting for training datasets where the loss can be minimized to near zero, which admits a *collocation* and *query-dependent partial orders*. A collocation is a set of token pairs where each token is directly paired with its subsequent token. Query-dependent partial orders is a set of partial orders where each partial order classifies tokens into three categories: optimal tokens, non-optimal tokens and non-comparable tokens. These structural properties define favorable max-margin problems on both the feed-forward layer and the self-attention layer.
- Second, we design a two-stage training algorithm based on normalized gradient descent. In stage 1 of pre-processing, we use the collocation to train the feed-forward layer. In stage 2, we use the entire dataset to train the self-attention layer. We show that the feed-forward layer and the query-key attention matrix converge sublinearly in direction respectively to the max-margin solution for classifying next token from all other tokens in the preprocessing dataset, and to the max-margin solution for classifying the optimal from non-optimal tokens. In addition, the norm of the transformer parameters grows linearly, which further yields a *linear* convergence rate of the cross-entropy loss. Our two-stage algorithm decouples the training of the feed-forward and attention layers without losing optimality, as stage 1's max-margin solution is judiciously designed to facilitate stage 2's fine-grained classification for optimal token prediction.
- Third, we show that the trained transformer has generalization ability for making non-trivial prediction on unseen data. In particular, the transformer is trained to learn an extended query-dependent partial order, where the non-comparable tokens are inserted in between the optimal tokens and non-optimal tokens. Thus, the trained transformer will attend to non-comparable tokens if optimal tokens are not in a new sentence and further make desirable prediction.

## 2   Related Work

Inspired by Brown et al. (2020), who demonstrated that pre-trained transformers can learn in-context - i.e., learn new tasks during inference with only a few samples - a series of works focus on the expressiveness power of transformers (Akyürek et al., 2022; Bai et al., 2023; Von Oswald et al., 2023; Fu et al., 2023; Giannou et al., 2023; Lin et al., 2023). These studies have shown that there exist parameter configurations such that transformers can perform various algorithms such as gradient descent. Additionally, Edelman et al. (2022) showed that transformers can represent a sparse function.

Regarding the training dynamics and optimization of transformers under in-context learning, Ahn et al. (2024); Mahankali et al. (2023); Zhang et al. (2023); Huang et al. (2023) studied the dynamics of a single attention layer, single-head transformer for the in-context learning of linear regression tasks. Cui et al. (2024) proved that multi-head attention outperforms single-head attention. Cheng et al. (2023) showed that local optimal solutions in transformers can perform gradient descent in-context for non-linear functions. Kim and Suzuki (2024) studied the nonconvex mean-field dynamics of transformers, and Nichani et al. (2024) established a convergence rate of $\tilde{O}(1/t)$ for the training loss in learning a causal graph. Additionally, Chen et al. (2024) investigated the gradient flow in training multi-head attention. Chen and Li (2024) proposed a supervised training algorithm for multi-head transformers.

Another line of research focuses on the training dynamics of transformers for binary classification problems. Tarzanagh et al. (2023b,a) demonstrated an equivalence between the optimization dynamics of a single attention layer and a certain SVM problem. While Tarzanagh et al. (2023b,a) only proved an asymptotic convergence result, Vasudeva et al. (2024) improved the convergence rate to $t^{-3/4}$. Li et al. (2023) studied the training dynamics of vision transformers and showed that the generalization error

can approach zero given sufficient training samples. Additionally, Deora et al. (2023) investigated the training and generalization error under the neural tangent kernel (NTK) regime.

For transformers trained on next-token prediction (NTP), Tian et al. (2023a) analyzed the training dynamics of a single-layer transformer, while Tian et al. (2023b) studied the joint training dynamics of multi-layer transformers. Li et al. (2024) demonstrated the asymptotic convergence of transformers trained with a logarithmic loss function for NTP. Although these works provided valuable insights into the training dynamics of transformers for NTP, they did not provide the finite-time convergence analysis, which is the focus of this paper. We remark that Thrampoulidis (2024) studied NTP without transformer structure.

Our work is also related to the classical implicit bias framework for training neural networks (NNs). In particular Soudry et al. (2018); Nacson et al. (2019); Ji and Telgarsky (2021); Ji et al. (2021) established convergence rate of gradient descent-based optimization. Phuong and Lampert (2020); Frei et al. (2022); Kou et al. (2024) studied the implicit bias of ReLU/Leaky-ReLU networks on orthogonal data. A comprehensive survey is provided in Vardi (2023). However, these works focused on classical neural networks, whereas we investigate the implicit bias of transformers for NTP.

## 3 Problem Setup

**Notations.** All vectors considered in this paper are column vectors. We use $\mathbb{1}\{A\}$ to denote the indicator function of $A$, i.e., $\mathbb{1}\{A\} = 1$ if $A$ holds, and $\mathbb{1}\{A\} = 0$ otherwise. $\|W\|$ represents the Frobenious norm of the matrix $W$. For a vector $v$, we use $[v]_i$ to denote the $i$-th coordinate of $v$. We use $\phi(v)$ to denote the softmax function, i.e., $[\phi(v)]_i = \exp(v_i)/\sum_j \exp(e_j^\top v)$, which can be applied to any vector with arbitrary dimension. We use $\{e_i\}_{i \in [|\mathcal{V}|]}$ to denote the canonical basis of $\mathbb{R}^{|\mathcal{V}|}$, i.e., $[e_i]_j = \mathbb{1}\{i = j\}$. The inner product $\langle A, B \rangle$ of two matrices $A, B$ equals to $\mathrm{Trace}(AB^\top)$.

**Next-token prediction.** We consider the task of next-token prediction, which aims to predict the subsequent token in a token sequence given its preceding tokens. Formally, suppose that there exists a finite vocabulary set $\mathcal{V} \subset \mathbb{R}^d$ that consists of all possible tokens, where $d$ is the dimension of the embedding. Each token $x \in \mathcal{V}$ is associated with a unique index $\mathrm{I}(x) \in \{1, 2, \ldots, |\mathcal{V}|\}$, where $\mathrm{I}$ is the index function. An $L$-length sentence $X = [x_1, \ldots, x_L] \in \mathcal{V}^L \subset \mathbb{R}^{d \times L}$ is a sequence of $L$ tokens, where $L$ is an integer. We assume that the maximum length of sentences is $L_{\max}$. The subsequent tokens in sentences are generated from a set of ground-truth model $\{p_L^* : \mathcal{V}^L \to \mathcal{V}\}_{L < L_{\max}}$, where $p_L^*$ generates the next token $x_{L+1}$ given the sentence $X$ for any $1 \leq L < L_{\max}$. The task of next-token prediction requires us to learn all models $\{p_L^*\}_{L < L_{\max}}$ given a training dataset $\mathcal{D}_0 = \{(X, x_{L+1})|L < L_{\max}, X \in \mathcal{V}^L, x_{L+1} \in \mathcal{V}\}$. Notably, if $X = [x_1, \ldots, x_L] \in \mathcal{D}_0$, then for any $\ell < L$, $([x_1, \ldots, x_\ell], x_{\ell+1})$ is also a training sample, since it follows $p_\ell^*$ as well.

**Decoder-only transformer.** A decoder-only transformer is a stack of blocks consisting of a self-attention layer and a feed-forward layer. For simplicity, we consider one-layer transformer, where the self-attention layer is determined by three matrices: $W_k \in \mathbb{R}^{d \times d_1}$, $W_q \in \mathbb{R}^{d_1 \times d}$ and $W_v \in \mathbb{R}^{d_2 \times d}$, namely key, query, and value matrices, and the feed-forward layer is determined by $W_o \in \mathbb{R}^{|\mathcal{V}| \times d_2}$. Here $d_1, d_2$ are hidden dimensions. Mathematically, given the input $X = [x_1 \ldots, x_L]$, we write the one-layer transformer as $\mathrm{T}_\theta(X) := \phi(W_o W_v X \phi(X^\top W_k W_q x_L)) \in [0, 1]^{|\mathcal{V}|}$, where $\theta := (W_o, W_v, W_k, W_q)$, and $\phi$ is the softmax function. We note that the inner softmax function $\phi$ is part of the attention model, and the outer softmax function $\phi$ is the decoder that generates a probability distribution over $\mathcal{V}$ for token prediction.

**Reparameterization.** We reparameterize the transformer architecture by consolidating the key and query matrices into a unified matrix $W_{kq}$, such that $W_{kq} = W_k W_q$. Similarly, we reparameterize the product of the feed-forward ($W_o$) and value ($W_v$) matrices as a single matrix $W_{ov}$, defined as $W_{ov} = W_o W_v$. Such a reparameterization is commonly adopted in transformer theory works (Huang et al., 2023; Tian et al., 2023a; Li et al., 2024; Nichani et al., 2024). Thus, the transformer under those reparameterization is given by $\mathrm{T}_\theta(X) := \phi(W_{ov} X \phi(X^\top W_{kq} x_L)) \in [0, 1]^{|\mathcal{V}|}$.

**Cross-entropy loss.** Given the training dataset $\mathcal{D}_0$ and the transformer model, we seek to learn $p_*$ by minimizing (training) the *cross-entropy* loss $\mathcal{L}(\theta)$ defined as follows:

$$\mathcal{L}(\theta) = -\frac{1}{|\mathcal{D}_0|} \sum_{(X, x_{L+1}) \in \mathcal{D}_0} \log e_{\mathrm{I}(x_{L+1})}^\top \mathrm{T}_\theta(X),$$

where $\mathrm{I}(x_{L+1})$ is the index of $x_{L+1}$ in $\mathcal{V}$.

# 4 Realizable Training Dataset and Two-Stage Algorithm

In this section, we first provide a mathematical framework based on partial order to formally characterize a realizable training dataset for next-token prediction. We will then describe a two-stage algorithm for next-token prediction that we study.

## 4.1 Realizable Training Dataset

We characterize a realizable training dataset via two structural properties, where the training loss can be made arbitrarily close to zero. We first provide some intuitions about those two properties.

**Existence of "collocation".** First, we note that if a sentence $X = [x_1, \ldots, x_L]$ is a legal training sample, $([x_1], x_2)$ is also in the training dataset. In addition, the output of a transformer given one single input token only depends on $W_{\mathrm{ov}}$, i.e. the feed-forward layer. Since training loss can be arbitrarily close to 0, there exists a sequence $\{W_t\}$ such that $\lim_{t \to \infty} - \sum_{x \in \mathcal{D}_0} \log e_\iota(x)^\top \phi(W_t x) = 0$, where $\iota(x)$ is the index of next token of $x$, and the summation is over the case when $x$ is the first token. Due to that $\phi(W_t x)$ is a probability distribution, the equality holds only when $\iota$ is injective, since otherwise it is an entropy of some distribution which is strictly greater than 0. Therefore, there exists an injective map $\mathrm{n} : \mathcal{V} \to \mathcal{V}$ such that every sentence starts with $x$, must have a unique next token $\mathrm{n}(x)$. We call the set of pairs $\{x, \mathrm{n}(x)\}_{x \in \mathcal{V}}$ a *collocation*. We remark that $p_1^* = \mathrm{n}$.

**Existence of "order".** Second, let us consider the output of a transformer $\mathrm{T}_\theta$ given a legal sentence $X = [x_1, \ldots, x_L]$ with the next token $x_{L+1} = p_L^*(X)$. The transformer first calculates a convex combination of $x_1, \ldots, x_L$ with corresponding weight $\varphi_\ell \propto \exp(x_\ell^\top W_{\mathrm{kq}} x_L)$ for each $\ell \leq L$. Then, the transformer outputs $\phi(\sum_\ell W_{\mathrm{ov}} x_\ell \varphi_\ell)$. Recall that the collocation forces $x_\ell$ to map to $\mathrm{n}(x_\ell)$, thus $\phi(W_{\mathrm{ov}} x_\ell)$ has a peak value at the coordinate equal to $\mathrm{I}(\mathrm{n}(x_\ell))$ (the index of $\mathrm{n}(x_\ell)$). Hence, $\mathrm{T}_\theta(X)$ can only have peak value at the coordinates within the set $\{\mathrm{I}(\mathrm{n}(x_\ell))\}_{\ell \leq L}$. If the training loss can be arbitrarily close to 0, it is desirable to have $\mathrm{n}^{-1}(x_{L+1}) \in \{x_\ell\}_{\ell \leq L}$. Therefore, for those $x_\ell$ with $\mathrm{n}(x_\ell) = x_{L+1}$, $\varphi_\ell$ must be larger than $\varphi_{\ell'}$ with $\mathrm{n}(x_{\ell'}) \neq x_{L+1}$. Finally, it worth noting that $\varphi_\ell$ depends on the final token $x_L$. This observation motivates us to define *query-dependent partial orders* on $\mathcal{V}$.

**Definition 1** ($x^q$-**partial order**) *Fix a token $x^q$. An $x^q$-partial order assigns an ordering relationship $>_{x^q}$ for certain pairs of tokens in $\mathcal{V}$, and is created as follows. Let $\mathcal{D}_0^{x^q}$ be the set of all legal sentences in the training dataset that has the final token (query) $x^q$. Then, for any pair of tokens $x, x' \in \mathcal{V}$, we assign $x >_{x^q} x'$ if there exists a sentence $X = [x_1, \ldots, x_L] \in \mathcal{D}_0^{x^q}$ and $x, x'$ are tokens in $X$ such that $\mathrm{n}(x) = x_{L+1} \neq \mathrm{n}(x')$, where $x_{L+1}$ is the next token of $X$.*

Note that Definition 1 is a "constructive definition" which might not be well-defined. However, as we are under the setting when the training loss can be arbitrarily close to 0, the aforementioned discussion shows that if $x >_{x^q} x'$, then $\varphi_\ell > \varphi_{\ell'}$, where $x = x_\ell$ and $x' = x_{\ell'}$ in some sentence. Thus, $\exp(x W_{\mathrm{kq}} x^q) > \exp(x' W_{\mathrm{kq}} x^q)$, which indeed need to be well-defined. Otherwise, we will have contradictions such as $\exp(x W_{\mathrm{kq}} x^q) > \exp(x' W_{\mathrm{kq}} x^q) < \exp(x W_{\mathrm{kq}} x^q)$. Mathematically, a well-defined (strict) partial order $>$ on a set $\mathcal{V}$ satisfies two axioms (Yannakakis, 1982): (i) there is no $x > x$; (ii) if $x > x'$ and $x' > x''$, then $x > x''$. Thus, $x^q$-partial order created by $\mathcal{D}_0$ is well-defined for every $x^q \in \mathcal{V}$.

Finally, let us discuss the impact of query-dependent partial orders on $\mathcal{D}_0$. For a given query $x^q$, the partial order $>_{x^q}$ divides tokens in $\mathcal{V}$ into four disjoint types.

- *(Strict) optimal tokens.* A token $x$ is optimal, if there is no $x'$ such that $x' >_{x^q} x$[1].
- *Confused tokens.* A token $x$ is confused, if there exists $x', x''$ such that $x' >_{x^q} x >_{x^q} x''$.
- *(Strict) non-optimal tokens.* A token $x$ is non-optimal if there is no $x'$ such that $x >_{x^q} x'$[2].
- *Non-comparable tokens.* A token $x$ is non-comparable if there is no $x'$ such that $x >_{x^q} x'$ or $x' >_{x^q} x$.

---

[1]This is also related to the maximal element in a partially ordered set.

[2]This is also related to the minimal element in a partially ordered set.

In this work, we assume that there are no confused tokens. This assumption simplifies the problem, making it tractable to provide explicit convergence in direction for training a transformer in Section 5. In summary, we make the following structural assumption on the training dataset.

**Assumption 1 (Realizable training dataset)** $\mathcal{D}_0$ *admits (i) a collocation* $\{x, \mathrm{n}(x)\}_{x \in \mathcal{V}}$*; (ii) well-defined query-dependent partial orders, where every* $x^q$*-partial order has no confused tokens.*

We remark that combining the collocation and query-dependent partial orders, we can regenerate the training dataset as follows. For any sentence with only one token $X = [x]$, the next token is $\mathrm{n}(x)$. For other sentences $X = [x_1, \ldots, x_L]$, let $x_\ell$ be optimal under the partial order $>_{x_L}$, and then the next token of $X$ is $\mathrm{n}(x_\ell)$. We next provide a simple example that justifies Assumption 1.

**Example 1** *Consider a language system where the vocabulary consists of four tokens* $\{S, V, O, P\}$*, where S,V,O,P respectively stand for subject, verb, object, and punctuation mark. This system admits the commonly adopted word order (Dryer, 1991): S, V, O, P. Let the training dataset be* $\{SVOP, VOP, OPP, PSV\}$*.*

*Let us create the corresponding collocation and the query-dependent partial orders from the dataset. The collocation is* $\{(S, V), (V, O), (O, P), (P, S)\}$*. That is, if a sentence starts with a subject, then the next token is a verb. Similarly, if a sentence starts with a verb, then the next token is an object, and so on. The query-dependent partial orders are created as follows:*

*Partial order under query S.* $S >_S P$.        *Partial order under query V.* $V >_V S$.
*Partial order under query O.* $O >_O S$, $O >_O V$.      *Partial order under query P.* $O >_P P$.

*Therefore, if a sentence starts with S (subject), the next token is V (verb) according to the collocation. Then, for the sentence SV, since the query is V and* $V >_V S$*, the next token of the sentence coincides with the next token of V, which is exactly O (object). Finally, for the sentence SVO, following similar argument, the next token is P (punctuation mark). This example satisfies Assumption 1 and aligns with real-world scenarios. An illustration is provided in Figure 1.*

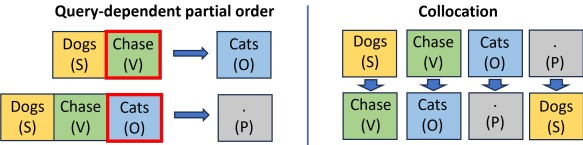

Figure 1: The left plot shows the mapping from sentence to the next token. The red rectangle indicates the optimal token in the corresponding sentence. The right plot shows the collocation relationship.

**Additional notations of training data.** It is worth noting that there are only finite number of distinct sentences. For ease of presentation, we introduce the following notations. Suppose there are $N$ distinct sentences in the training dataset $\mathcal{D}_0$ indexed by $n \in \{1, \ldots, N\}$. For each distinct sentence $X^{(n)}$, we calculate its frequency $\pi^{(n)} \in [0, 1]$ in dataset $\mathcal{D}_0$ as $\pi^{(n)} = \frac{\sum_{(X, x_{L+1}) \in \mathcal{D}_0} \mathbb{1}\{X = X^{(n)}\}}{|\mathcal{D}_0|}$.

Building upon this, with a little abuse of notation, we use $\mathrm{n}(X^{(n)}) \in \mathcal{V}$ to denote the subsequent token of the sentence $X^{(n)}$ and $\mathrm{In}(X^{(n)})$ to denote the index of $\mathrm{n}(X^{(n)})$.

We further denote $X_{-1}^{(n)}$ as the final token of $X^{(n)}$, and let $\bar{\mathrm{T}}_\theta(X) = W_{\mathrm{ov}} X \phi(X^\top W_{\mathrm{kq}} X_{-1}^{(n)})$. Then, the loss function $\mathcal{L}(\theta)$ can be rewritten as follows:

$$\mathcal{L}(\theta) = \sum_n \pi^{(n)} \left( \log \left( \sum_v \exp \left( e_v^\top \bar{\mathrm{T}}_\theta(X^{(n)}) \right) \right) - e_{\mathrm{In}(X^{(n)})}^\top \bar{\mathrm{T}}_\theta(X^{(n)}) \right). \tag{1}$$

## 4.2 Training Algorithm

For the realizable dataset satisfying Assumption 1, we propose a two-stage training algorithm using normalized gradient descent (NGD). The pseudo code of the algorithm is presented in Algorithm 1. In Section 5, we show that the two-stage algorithm decouples the training of the feed-forward and

attention layers without losing the optimality. This is because the training in stage 1 is designed to yield a suitable max-margin solution, which will enable the training of stage 2 to solve a fine-grained classifcation problem and identify the optimal token for prediction.

In the first stage of pre-processing, we use the collocation set to train the feed-forward layer $W_{\mathrm{ov}}$. For simplicity, we introduce the following notation for the training loss of the feed-forward layer. Given a collocation $\{x, \mathrm{n}(x)\}_{x \in \mathcal{V}}$, which can be obtained through extracting all length-2 sentences in the training dataset $\mathcal{D}_0$[3], we use normalized gradient descent to train $W_{\mathrm{ov}}$. Equivalently, the loss function can be written as

$$\mathcal{L}_0(W_{\mathrm{ov}}) = -\sum_{x \in \mathcal{V}} \log \frac{\exp(e_{\mathrm{In}(x)}^\top W_{\mathrm{ov}} x)}{\sum_{v \leq |\mathcal{V}|} \exp(e_v^\top W_{\mathrm{ov}} x)},$$

where the self-attention elements are removed because the attention matrices are not trained here. Based on the above loss function, we initialize $W_{\mathrm{ov}}^{(0)} = 0 \in \mathbb{R}^{|\mathcal{V}| \times d}$, and subsequently take an update at each time $t$ by NGD as in line 4 of Algorithm 1.

In the second stage, we fix the trained feed-forward layer and train the self-attention layer based on the loss function given in Equation (1) and using the entire dataset $\mathcal{D}_0$. Specifically, we initialize $W_{\mathrm{kq}} = 0 \in \mathbb{R}^{d \times d}$, and subsequently take an update at each time $t$ by NGD as in line 7 of Algorithm 1.

---

**Algorithm 1** Two-stage Normalized Gradient Descent

---

1: **Initialization:** $W_{\mathrm{ov}}^{(0)} = 0 \in \mathbb{R}^{|\mathcal{V}| \times d}$, $W_{\mathrm{kq}} = 0 \in \mathbb{R}^{d \times d}$.
2: **Input:** A collocation $\{x, \mathrm{n}(x)\}_{x \in \mathcal{V}}$, and a training dataset $\mathcal{D}_0$, learning rate $\eta_0, \eta$.
3: **for** $t \in \{0, 1, ..., T-1\}$ **do**
4:     Update $W_{\mathrm{ov}}^{(t+1)}$ as $W_{\mathrm{ov}}^{(t+1)} = W_{\mathrm{ov}}^{(t)} - \eta_0 \frac{\nabla_{W_{\mathrm{ov}}} \mathcal{L}_0(W_{\mathrm{ov}}^{(t)})}{\|\nabla_{W_{\mathrm{ov}}} \mathcal{L}_0(W_{\mathrm{ov}}^{(t)})\|}$.
5: **end for**
6: **for** $t \in \{0, \ldots, T_1 - 1\}$ **do**
7:     Update $W_{\mathrm{kq}}^{(t+1)}$ as $W_{\mathrm{kq}}^{(t+1)} = W_{\mathrm{kq}}^{(t)} - \eta \frac{\nabla_{W_{\mathrm{kq}}} \mathcal{L}(\theta^{(t)})}{\|\nabla_{W_{\mathrm{kq}}} \mathcal{L}(\theta^{(t)})\|}$, where $\theta^{(t)} = (W_{\mathrm{ov}}^{(T)}, W_{\mathrm{kq}}^{(t)})$.
8: **end for**

---

## 5 Training Dynamics of the Transformer

In this section, we present the convergence result for Algorithm 1. Before we proceed, we first introduce the following technical assumption, which has been commonly adopted in the previous theoretical studies of transformers (Huang et al., 2023; Li et al., 2024; Tian et al., 2023a).

**Assumption 2** *The vocabulary set is orthornormal. Namely, the embedding has unit norm, i.e., $\|x\| = 1$, and $x^\top x' = 0$ holds for any distinct tokens $x$ and $x'$.*

### 5.1 Convergence of Training $W_{\mathrm{ov}}$

To characterize the training dynamics of $W_{\mathrm{ov}}$, we observe that the collocation $\{(x, \mathrm{n}(x))\}_{x \in \mathcal{V}}$ defines the following hard-margin problem:

$$W_{\mathrm{ov}}^* = \arg\min \|W\|, \quad \text{s.t.} \quad (e_{v^*} - e_v) W x \geq 1, \quad \forall v^* = \mathrm{In}(x), v \neq \mathrm{In}(x). \quad (2)$$

It can be shown that $\lim_{B \to +\infty} \mathcal{L}_0(B W_{\mathrm{ov}}^*) = 0$. Thus, the loss function $\mathcal{L}_0$ trains $W_{\mathrm{ov}}$ to be the max-margin solution with $W_{\mathrm{ov}} x$ distinguishing the next token $\mathrm{n}(x)$ from all other tokens in $\mathcal{V}$.

Since $\mathcal{L}_0(\cdot)$ is convex, we have the following convergence result on the training of $W_{\mathrm{ov}}^{(t)}$.

**Proposition 1** *Let $W_{\mathrm{ov}}^*$ be defined in Equation (2). Under Assumptions 1-2, let $W_{\mathrm{ov}}^{(t)}$ be updated by Algorithm 1. Then, for any $t \geq 2$, we have $\frac{t\eta_0}{2\|W_{\mathrm{ov}}^*\|} \leq \|W_{\mathrm{ov}}^{(t)}\| \leq t\eta_0$ and the following bound holds:*

---
[3]A more general way is to use various standard techniques developed in linguistic analysis (Lehecka, 2015).

$$\left\langle \frac{W_{\text{ov}}^{(t)}}{\|W_{\text{ov}}^{(t)}\|}, \frac{W_{\text{ov}}^*}{\|W_{\text{ov}}^*\|} \right\rangle \geq 1 - \frac{5\|W_{\text{ov}}^*\|^3 \log(2|\mathcal{V}|) \log t}{t\eta_0}.$$

*Moreover, the loss function $\mathcal{L}_0$ satisfies that $\mathcal{L}_0(W_{\text{ov}}^{(t)}) \leq O(\exp(-\eta_0 t/(4\|W_{\text{ov}}^*\|)))$.*

Proposition 1 states that during the training stage 1, the feed-forward layer $W_{\text{ov}}^{(t)}$ converges in direction to $W_{\text{ov}}^*/\|W_{\text{ov}}^*\|$ at a rate of $O(\log t/t)$, which classifies the next token from all other tokens. In addition, since the norm of $W_{\text{ov}}^{(t)}$ increases linearly, the loss $\mathcal{L}_0(W_{\text{ov}}^{(t)})$ converges linearly to zero, i.e., $\mathcal{L}_0(W_{\text{ov}}^{(t)}) = O(\exp(-C_0 t))$ for some constant $C_0$.

## 5.2 Convergence of Training $W_{\text{kq}}$

Recall that after the training stage 1 with $T$ steps, we obtain a trained feed-forward layer $W_{\text{ov}}^{(T)}$. Then, we fix $W_{\text{ov}}^{(T)}$ and use normalized gradient descent to train $W_{\text{kq}}$. To characterize the training dynamics of the key-query matrix $W_{\text{kq}}$, we note that each query-dependent partial order also defines a hard-margin problem. Let $l(n) \subset \{1, \dots, L^{(n)}\}$ be the set of indices of the optimal tokens of $X^{(n)}$. Recall that $x_\ell$ is optimal if there is no $x_{\ell'}$ such that $x_{\ell'} >_{X_{-1}^{(n)}} x_\ell$ and $\text{In}(x_\ell) = \text{In}(X^{(n)})$. That is, $W_{\text{kq}} X_{-1}^{(n)}$ should correctly classify optimal tokens $x_\ell$ and non-optimal tokens $x_{\ell'}$. This is formalized in the following problem:

$$W_{\text{kq}}^* = \arg\min \|W\|, \quad \text{s.t.} \quad (x_{\ell_*}^{(n)} - x_\ell^{(n)})W X_{-1}^{(n)} \geq 1, \quad \forall \ell_* \in l(n), \ell \notin l(n), \forall n. \quad (3)$$

We will show that the loss function in Equation (1) given the well trained $W_{\text{ov}}^{(T)}$ will train $W_{\text{kq}}$ towards the max-margin solution $W_{\text{kq}}^*$ in direction for classifying between the optimal and non-optimal token. We further make the following technical assumption.

**Assumption 3** *For any sample $X^{(n)}$, the number of optimal tokens is not less than the number of non-optimal tokens. Formally, for any non-optimal token $x$ in $X^{(n)}$, we have $|l(n)| \geq \sum_\ell \mathbb{1}\{x_\ell = x\}$.*

Assumption 3 is consistent with practical and empirical observations, where optimal tokens often demonstrate higher relevance, making them more frequent in subsequent outcomes.

We now present the convergence result for the training of the key-query matrix in stage 2.

**Theorem 1** *Let Assumptions 1-3 hold. Let $W_{\text{kq}}^*$ be the solution of Equation (3). Let $\eta < O(1)$ and $W_{\text{kq}}^{(t)}$ be updated by Algorithm 1. Then, for any $t \geq 2$, we have that $\frac{t\eta}{2\|W_{\text{kq}}^*\|} \leq \|W_{\text{kq}}^{(t)}\| \leq t\eta$. In addition, the following inequality holds.*

$$\left\langle \frac{W_{\text{kq}}^{(t)}}{\|W_{\text{kq}}^{(t)}\|}, \frac{W_{\text{kq}}^*}{\|W_{\text{kq}}^*\|} \right\rangle \geq 1 - \frac{54NL_{\max}^4\|W_{\text{kq}}^*\|^4 \log^2 t}{t\eta}.$$

Theorem 1 states that the key-query matrix $W_{\text{kq}}$ converges in direction to the max-margin solution $W_{\text{kq}}^*/\|W_{\text{kq}}^*\|$ at a convergence rate of $O(\log^2 t/t)$. We further show that the norm of $W_{\text{kq}}$ also grows linearly in $t$, i.e., $\|W_{\text{kq}}\| = \Omega(t)$. Combining these results, we have the following theorem on the convergence of the loss function and the training accuracy.

**Theorem 2 (Loss Convergence)** *For any training sentence $X^{(n)} = [x_1^{(n)}, \dots, x_L^{(n)}]$, let $\varphi_\ell^{(n,t)} \propto \exp(x_\ell^{(n)} W_{\text{kq}}^{(t)} x_L^{(n)})$ be the attention weight. Under the conditions in Theorem 1, there is an absolute constant $c_0$ such that when $T \geq c_0\|W_{\text{ov}}^*\|^5 \log(|\mathcal{V}|) \log T/\eta_0$ and $t \geq c_0 N L_{\max}^4 \|W_{\text{kq}}^*\|^6 \log^2 t/\eta$, the optimal token weight satisfies $\min_n \sum_{\ell_* \in l(n)} \varphi_{\ell_*}^{(n,t)} \geq (1 + L_{\max} \exp(-tC_1))^{-1}$. In addition, the loss function $\mathcal{L}$ converges linearly[4] to its minimal value:*

$$\mathcal{L}(\theta^{(t)}) = \mathcal{L}(W_{\text{ov}}^{(T)}, W_{\text{kq}}^{(t)}) \leq |\mathcal{V}| \exp\left(-TC_0\left(1 - \frac{2L_{\max}}{L_{\max} + \exp(C_1 t)}\right)\right), \quad (4)$$

---

[4]For fixed $W_{\text{ov}}^{(T)}$, the minimum loss value $\mathcal{L}^* = |\mathcal{V}| \exp(-TC_0)$. Then Equation (4) implies $\mathcal{L}(\theta^{(t)}) - \mathcal{L}^* \leq \mathcal{L}^* T C_0 O(e^{-C_1 t})$ for sufficiently large $t$, which further implies the linear convergence in $t$.

where $C_0 = \frac{\eta_0}{4\|W_{\mathrm{ov}}^*\|^2}$ and $C_1 = \eta/(4L_{\max}\|W_{\mathrm{kq}}^*\|^2)$.

Theorem 2 shows that the training loss converges to its minimum value at a linear convergence rate. Furtherm, for $T = \Omega(\log(1/\epsilon_0))$ $t = \Omega(\log(1/\epsilon))$, the optimal token weight is given by $1/(1+\epsilon)$ for any $\epsilon > 0$, which is close to 1. This implies that the trained transformer attends to the optimal token and thus outputs the correct next token $\mathrm{n}(x_{\ell_*}^{(n)})$ with probability $1 - O(\epsilon_0)$.

### 5.3 Proof Sketch of Theorem 1

The proof consists of the following three main steps. The key proof step lies in carefully analyzing the projection of gradient $\nabla_{W_{\mathrm{kq}}}\mathcal{L}(\theta^{(t)})$ onto the token-query outer product $x_\ell^{(n)}(X_{-1}^{(n)})^\top$, max-margin attention weight matrix $W_{\mathrm{kq}}^*$, and the trained attention weight matrix $W_{\mathrm{kq}}^{(t)}$.

**Step 1 (Lemma 5).** By analyzing $\left\langle \nabla_{W_{\mathrm{kq}}}\mathcal{L}(\theta^{(t)}), x_\ell^{(n)}(X_{-1}^{(n)})^\top \right\rangle$, we characterize the dynamics of attention weights. Using mathematical induction, we show that the lower bound of optimal token weight is $1/L_{\max}$.

**Step 2 (Lemma 6).** Then, we show that the cosine similarity between the negative gradient and $W_{\mathrm{kq}}^*$ is strictly larger than the minimum optimal token weight. Utilizing step 1, due to the NGD update, the norm of the key-query matrix $W_{\mathrm{kq}}^{(t)}$ can be shown to grow linearly.

**Step 3 (Lemma 7).** Finally, we carefully compare the difference between the projections from gradient to the trained attention matrix and max-margin attention matrix. By separately evaluating the impact of the optimal and non-optimal tokens on those projections, we can show the following inequality for some constant $C_0$:

$$\left\langle \nabla_{W_{\mathrm{kq}}}\mathcal{L}(\theta^{(t)}), W_{\mathrm{kq}}^{(t)} \right\rangle \geq \left(1 + \frac{C_0 \log \|W_{\mathrm{kq}}^{(t)}\|}{\|W_{\mathrm{kq}}^{(t)}\|}\right) \left\langle \nabla_{W_{\mathrm{kq}}}\mathcal{L}(\theta^{(t)}), W_{\mathrm{kq}}^* \right\rangle \frac{\|W_{\mathrm{kq}}^{(t)}\|}{\|W_{\mathrm{kq}}^*\|}.$$

Utilizing step 2's result that $\|W_{\mathrm{kq}}^{(t)}\|$ grows linearly, the dynamics of the attention layer can be shown to converge in direction to the max-margin solution in Equation (3).

## 6 Generalization Ability

In this section, we prove the generalization ability of the trained transformers. Recall that Theorem 1 shows that $W_{\mathrm{kq}}^{(t)}$ converges to $W_{\mathrm{kq}}^*\|W_{\mathrm{kq}}^{(t)}\|/\|W_{\mathrm{kq}}^*\|$. To characterize the generalization ability, it is desirable to use the property of $W_{\mathrm{kq}}^*$, which is given in the following result.

**Proposition 2** *Under Assumptions 1-2, fix a query token $x^q$, let $\mathcal{O}_{x^q}, \mathcal{N}_{x^q}, \mathcal{M}_{x^q} \subset \mathcal{V}$ be the set of optimal tokens, the set of non-optimal tokens, and the set of non-comparable tokens, under $x^q$-partial order, respectively. Then, the solution $W_{\mathrm{kq}}^*$ of Equation (3) satisfies $x_0^\top W_{\mathrm{kq}}^* x^q = 0$ for $x_0 \in \mathcal{M}_{x^q}$, and*

$$x_*^\top W_{\mathrm{kq}}^* x^q = \frac{|\mathcal{N}_{x^q}|}{|\mathcal{O}_{x^q}| + |\mathcal{N}_{x^q}|}, \quad x^\top W_{\mathrm{kq}}^* x^q = -\frac{|\mathcal{O}_{x^q}|}{|\mathcal{O}_{x^q}| + |\mathcal{N}_{x^q}|}, \quad \forall x_* \in \mathcal{O}_{x^q}, x \in \mathcal{N}_{x^q}.$$

Recall that non-comparable tokens (see Section 4) under a query $x^q$ never appears in any training sentence data with the same query $x^q$. Thus, Proposition 2 implies an interesting generalization capability – each $x^q$-partial order can automatically incorporate more relationships to expand the query-dependent partial orders. Combining Proposition 2 with Theorem 1, we obtain the following theorem on $W_{\mathrm{kq}}^{(t)}$.

**Theorem 3** *Under the conditions and notations in Proposition 2, let $T = \Omega(\log(1/\epsilon))$, and $t = \Omega(\log(1/\epsilon))$. Then there exists a constant $C_0$ such that*

$$(x_* - x_0)^\top W_{\mathrm{kq}}^{(t)} x^q \geq C_0 t, \quad (x_0 - x)^\top W_{\mathrm{kq}}^{(t)} x^q \geq C_0 t, \quad \forall x_* \in \mathcal{O}_{x^q}, x_0 \in \mathcal{M}_{x^q}, x \in \mathcal{N}_{x^q}.$$

*Moreover, if the trained transformer takes input $X$ with query $x^q$ that consists of a non-comparable token $x_0$ and non-optimal tokens, then the prediction made by $\mathrm{T}_{\theta^{(t)}}(X)$ is $\mathrm{n}(x_0)$ with high probability.*

Theorem 3 suggests that a new partial order is created by the trained transformer. Specifically, it inserts the non-comparable tokens between the optimal and non-optimal tokens. The trained transformer can generalize the token prediction to such new sentences as given in Theorem 3.

We use Example 1 to illustrate the generalization ability described above.

**Example 2 (Generalization to unseen data in Example 1)** *Recall that in Example 1, the training dataset consists of four sentences: SVOP, VOP, OPP, and PSV. Consider the partial order $>_P$ under the punctuation mark P. We have that $O>_P P$ and O is an optimal token, P is a non-optimal token, and S,V are non-comparable tokens. We then have the following non-trivial prediction by the trained transformer.*

**Case 1.** *Non-comparable tokens are learned to be "larger" than non-optimal tokens.*

*Consider a new (unseen) input sentence SP. Since S is non-comparable before training, but is "larger" than P under the trained key-query matrix $W_{kq}^{(t)}$, the next predicted token is $n(S) = V$.*

**Case 2.** *Optimal tokens remain optimal over all tokens after training.*

*Consider a new (unseen) input OSP. O is optimal and S is still "smaller" than O under the trained P-partial order. The trained transformer will consistently predict P.*

In both of the above cases, the trained transformer provides desirable prediction for the unseen sentences. We further note that the effectiveness of both cases can vary during the inference time of the trained transformer. For instance, if the input sequence is SP (subject-punctuation), the output is SPV (subject-P-verb), which follows a logical subject-verb order and is desirable. However, in cases where the input is VP (verb-punctuation), it may be preferable to terminate the sequence after the verb, i.e., VPP, as the verb alone can suffice to convey the intended meaning.

# 7 Experiment

In this section, we verify our theoretical findings via an experiment on a synthetic dataset. Specifically, we randomly generate a realizable dataset as described in Assumption 1 with $|\mathcal{V}| = 20$. Then, we train $W_{ov}$ and $W_{kq}$ by Algorithm 1, each with 900 iterations. The parameters are chosen as $d = |\mathcal{V}|$, $\eta_0 = 0.2/\sqrt{d}$, and $\eta = 0.05/\sqrt{d}$. In Figure 2, the first three plots show the dynamics of the training stage 1, which indicates the convergence of the loss $\mathcal{L}_0(W_{ov}^{(t)})$ to its minimum value, the convergence of $W_{ov}^{(t)}$ in direction to $W_{ov}^*$, and the linear increase of the norm $\|W_{ov}^{(t)}\|$, respectively. These results verify Proposition 1. The last three plots show the dynamics of the training stage 2, which indicates the convergence of the loss $\mathcal{L}(\theta^{(t)})$, the convergence of $W_{kq}^{(t)}$ in direction to $W_{kq}^*$, and the linear increase of the norm $\|W_{kq}^{(t)}\|$. These results verify Theorem 1 and Theorem 2. All experiments are conducted on a PC equipped with an i5-12400F processor and 16GB of memory.

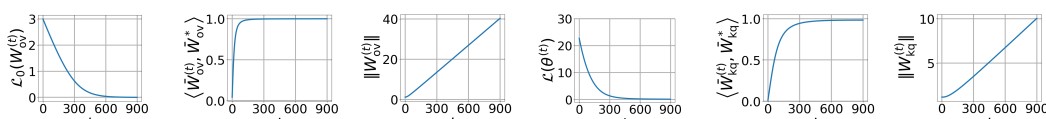

Figure 2: Training dynamics of single-layer transformer for NTP.

# 8 Conclusion

In this work, we investigated the training dynamics of a single-layer transformer for NTP. We first characterized two structural properties of the training dataset under the realizable setting where the training loss can be made arbitrarily close to zero. These properties allow us to define two max-margin solutions for both the feed-forward layer and the self-attention layer. Then, we showed that both layers converge in direction to their corresponding max-margin solutions sub-linearly, which further yields a linear convergence of the training loss for NTP. We further showed that the well trained transformer can have non-trivial prediction ability on unseen data, which sheds light on the generalization capability of transformers. Our experiments verify our theoretical findings.

## Acknowledgments and Disclosure of Funding

The work of R. Huang and J. Yang was supported in part by the U.S. National Science Foundation under grants NSF CNS-1956276 and ECCS-2133170. The work of Y. Liang was supported in part by the U.S. National Science Foundation under grants ECCS- 2413528 and DMS-2134145.

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

# Contents

# A    Expression of Gradients

We first provide the general formula for the gradients of both layers.

$$\nabla_{W_{\text{ov}}} \mathcal{L}_0(W_{\text{ov}}) = \sum_{x \in \mathcal{V}} \left( \text{T}_0(x) - e_{\text{In}(x)} \right) x^\top, \tag{5}$$

$$\nabla_{W_{\text{kq}}} \mathcal{L}(\theta)$$
$$= \sum_n \pi^{(n)} X^{(n)} \left( \text{diag}(\phi_\theta(X^{(n)})) - \phi_\theta(X^{(n)})\phi_\theta(X^{(n)})^\top \right) (X^{(n)})^\top W_{\text{ov}}^\top \left( \text{T}_\theta(X^{(n)}) - p^{(n)} \right) (X_{-1}^{(n)})^\top. \tag{6}$$

# B    Proof of Proposition 1

Recall that we use the loss,

$$\mathcal{L}_0(W_{\text{ov}}) = -\sum_{x \in \mathcal{V}} \log \frac{\exp\left( e_{\text{In}(x)}^\top W_{\text{ov}} x \right)}{\sum_{i \in [|\mathcal{V}|]} \exp\left( e_i^\top W_{\text{ov}} x \right)}.$$

The updating rule of $W_{\text{ov}}$ is that

$$W_{\text{ov}}^{(t+1)} = W_{\text{ov}}^{(t)} - \eta_0 \frac{\nabla_{W_{\text{ov}}} \mathcal{L}_0(W_{\text{ov}}^{(t)})}{\|\nabla_{W_{\text{ov}}} \mathcal{L}_0(W_{\text{ov}}^{(t)})\|}. \tag{7}$$

We know that $\mathcal{L}_0$ is convex respect to $W_{\text{ov}}$. Therefore, we have

$$\left\langle W_{\text{ov}}^{(t)} - W_{\text{ov}}', \nabla_{W_{\text{ov}}} \mathcal{L}_0(\theta^{(t)}) \right\rangle \geq \mathcal{L}_0(\theta^{(t)}) - \mathcal{L}_0(\theta').$$

It is clear that the loss function $\mathcal{L}$ reaches the minimum 0 when $W_{\text{ov}} = \Delta W_{\text{ov}}^*$, as $\Delta \to \infty$.

**Lemma 1** *Under the initialization $W_{\text{ov}}^{(0)}$ and the updating rule Equation (7) with step size $\eta$, the following inequality holds.*

$$t\eta_0 + \|W_{\text{ov}}^{(0)}\| \geq \|W_{\text{ov}}^{(t)}\| \geq \frac{t\eta_0}{2\|W_{\text{ov}}^*\|} - \|W_{\text{ov}}^{(0)}\|.$$

*Proof.*  Using Equation (5), we have

$$\left\langle W_{\text{ov}}^*, \nabla_{W_{\text{ov}}} \mathcal{L}_0(W_{\text{ov}}^{(t)}) \right\rangle = \sum_{x \in \mathcal{V}} \left( \text{T}_0^{(t)}(x) - e_{\text{In}(x)} \right)^\top W_{\text{ov}}^* x$$

$$= \sum_{x \in \mathcal{V}} \sum_{i \in [|\mathcal{V}|]} [\text{T}_0^{(t)}(x)]_i (e_i - e_{\text{In}(x)})^\top W_{\text{ov}}^* x$$

$$\overset{(a)}{\leq} -\sum_{x \in \mathcal{V}} \sum_{i \neq \text{In}(x)} [\text{T}_0^{(t)}(x)]_i, \tag{8}$$

where $(a)$ is due the constraints that $W_{\text{ov}}^*$ satisfies. On the other hand,

$$\|\nabla_{W_{\text{ov}}} \mathcal{L}(W_{\text{ov}}^{(t)})\| = \left\langle \frac{\nabla_{W_{\text{ov}}} \mathcal{L}(W_{\text{ov}}^{(t)})}{\|\nabla_{W_{\text{ov}}} \mathcal{L}(W_{\text{ov}}^{(t)})\|}, \nabla_{W_{\text{ov}}} \mathcal{L}_0(W_{\text{ov}}^{(t)}) \right\rangle$$

$$= \sum_{x \in \mathcal{V}} \sum_i [\mathrm{T}_0^{(t)}(x)]_i (e_i - e_{\mathrm{In}(x)})^\top \frac{\nabla_{W_{\mathrm{ov}}} \mathcal{L}(\theta^{(t)})}{\|\nabla_{W_{\mathrm{ov}}} \mathcal{L}(\theta^{(t)})\|} x$$

$$\overset{(a)}{\leq} 2 \sum_{x \in \mathcal{V}} \sum_{i \neq \mathrm{In}(x)} [\mathrm{T}_0^{(t)}(x)]_i,$$

where $(a)$ follows from $\|AB\| \leq \|A\|\|B\|$ for any matrices $A$ and $B$. Thus, we obtain

$$\left\langle W_{\mathrm{ov}}^*, \frac{\nabla_{W_{\mathrm{ov}}} \mathcal{L}(\theta^{(t)})}{\|\nabla_{W_{\mathrm{ov}}} \mathcal{L}(\theta^{(t)})\|} \right\rangle \leq -1/2$$

To lower bound the norm of $W_{\mathrm{ov}}$, we recall the updating rule (Equation (7)).

$$\|W_{\mathrm{ov}}^{(t)}\| = \left\| W_{\mathrm{ov}}^{(0)} - \sum_{t' < t} \eta_0 \frac{\nabla_{W_{\mathrm{ov}}} \mathcal{L}_0(W_{\mathrm{ov}}^{(t')})}{\|\nabla_{W_{\mathrm{ov}}} \mathcal{L}_0(W_{\mathrm{ov}}^{(t')})\|} \right\|$$

$$\geq \left\langle W_{\mathrm{ov}}^{(0)} - \sum_{t' < t} \eta_0 \frac{\nabla_{W_{\mathrm{ov}}} \mathcal{L}_0(W_{\mathrm{ov}}^{(t')})}{\|\nabla_{W_{\mathrm{ov}}} \mathcal{L}_0(W_{\mathrm{ov}}^{(t')})\|}, \frac{W_{\mathrm{ov}}}{\|W_{\mathrm{ov}}^*\|} \right\rangle$$

$$\geq \frac{t \eta_0}{2\|W_{\mathrm{ov}}^*\|} - \|W_{\mathrm{ov}}^{(0)}\|.$$

For the LHS of the inequality, it suffices to note that at each iteration, the norm $\|W_{\mathrm{ov}}^{(t)}\|$ increases most $\eta_0$ due to normalized gradient descent. ∎

**Lemma 2** *At each iteration $t$, the following inequality holds.*

$$\left\langle \frac{W_{\mathrm{ov}}^{(t)}}{\|W_{\mathrm{ov}}^{(t)}\|}, \nabla_{W_{\mathrm{ov}}} \mathcal{L}_0(W_{\mathrm{ov}}^{(t)}) \right\rangle \geq \left( 1 + \frac{2\|W_{\mathrm{ov}}^*\|}{\|W_{\mathrm{ov}}^{(t)}\|} \log(2|\mathcal{V}|) \right) \left\langle \frac{W_{\mathrm{ov}}^*}{\|W_{\mathrm{ov}}^*\|}, \nabla_{W_{\mathrm{ov}}} \mathcal{L}_0(W_{\mathrm{ov}}^{(t)}) \right\rangle$$

*Proof.* First, we consider the case when $W_{\mathrm{ov}}^{(t)} = W_{\mathrm{ov}}^* \frac{\|W_{\mathrm{ov}}^{(t)}\|}{\|W_{\mathrm{ov}}^*\|}$. Due to Equation (8) in Lemma 1, we have

$$\left\langle \frac{W_{\mathrm{ov}}^*}{\|W_{\mathrm{ov}}^*\|}, \nabla_{W_{\mathrm{ov}}} \mathcal{L}_0(W_{\mathrm{ov}}^{(t)}) \right\rangle < 0$$

In this case, the result is trivial.

Then, we consider the case when $W_{\mathrm{ov}}^{(t)} \neq W_{\mathrm{ov}}^* \frac{\|W_{\mathrm{ov}}^{(t)}\|}{\|W_{\mathrm{ov}}^*\|}$. Due the optimality of $W_{\mathrm{ov}}^*$, which achieves the minimum norm satisfying the constraints in Equation (2), we must have that for some $x_0 \in \mathcal{V}$, there exists $i_0 \neq \mathrm{In}(x_0)$ such that the following inequality holds

$$(e_{\mathrm{In}(x_0)} - e_i)^\top W_{\mathrm{ov}}^{(t)} x_0 < \frac{\|W_{\mathrm{ov}}^{(t)}\|}{\|W_{\mathrm{ov}}^*\|}.$$

Therefore, the loss on $W_{\mathrm{ov}}^{(t)}$ can be lower bounded as follows.

$$\mathcal{L}(W_{\mathrm{ov}}^{(t)}) = \sum_{x \in \mathcal{V}} \log \left( 1 + \sum_i \exp \left( (e_i - e_{\mathrm{In}(x)})^\top W_{\mathrm{ov}}^{(t)} x \right) \right)$$

$$> \log \left( 1 + \exp \left( -\|W_{\mathrm{ov}}^{(t)}\| / \|W_{\mathrm{ov}}^*\| \right) \right)$$

$$\overset{(a)}{>} \frac{1}{2} \exp \left( -\|W_{\mathrm{ov}}^{(t)}\| / \|W_{\mathrm{ov}}^*\| \right),$$

where $(a)$ is due to the fact that $\log(1 + x) \geq x/2$ when $0 < x < 1$. On the other hand, let $W_{\mathrm{ov}}' = \left( \frac{\|W_{\mathrm{ov}}^{(t)}\|}{\|W_{\mathrm{ov}}^*\|} + 2 \log(2|\mathcal{V}|) \right) W_{\mathrm{ov}}^*$. Then, the loss on $W_{\mathrm{ov}}'$ has the following upper bound.

$$\mathcal{L}(W'_{\text{ov}}) = \sum_{x \in \mathcal{V}} \log \left( 1 + \sum_i \exp \left( (e_i - e_{\text{In}(x)})^\top W_{\text{ov}}^{(t)} x \right) \right)$$

$$\leq \sum_{x \in \mathcal{V}} \log \left( 1 + (|\mathcal{V}| - 1) \exp \left( -\|W_{\text{ov}}^{(t)}\| / \|W_{\text{ov}}^*\| - \log(2|\mathcal{V}|) \right) \right)$$

$$\overset{(a)}{\leq} \sum_{x \in \mathcal{V}} |\mathcal{V}| \exp \left( -\|W_{\text{ov}}^{(t)}\| / \|W_{\text{ov}}^*\| - 2\log(2|\mathcal{V}|) \right)$$

$$\leq \frac{1}{2} \exp(-\|W_{\text{ov}}^{(t)}\| / \|W_{\text{ov}}^*\|),$$

where $(a)$ is due to the fact that $\log(1 + x) < x$ when $x > 0$. Thus, $\mathcal{L}(W_{\text{ov}}^{(t)}) > \mathcal{L}(W'_{\text{ov}})$. Due to the convextiy of $\mathcal{L}_0$, we have

$$0 < \left\langle W_{\text{ov}}^{(t)} - W'_{\text{ov}}, \nabla_{W_{\text{ov}}} \mathcal{L}_0(W_{\text{ov}}^{(t)}) \right\rangle$$

$$= \left\langle W_{\text{ov}}^{(t)}, \nabla_{W_{\text{ov}}} \mathcal{L}_0(W_{\text{ov}}^{(t)}) \right\rangle - \left( \frac{\|W_{\text{ov}}^{(t)}\|}{\|W_{\text{ov}}^*\|} + 2\log(2|\mathcal{V}|) \right) \left\langle W_{\text{ov}}^*, \nabla_{W_{\text{ov}}} \mathcal{L}_0(W_{\text{ov}}^{(t)}) \right\rangle,$$

which finishes the proof.

■

**Proposition 3 (Restatement of Proposition 1)** *Under the zero initialization $W_{\text{ov}}^{(0)} = 0$ and updating rule Equation (7), for any $t \geq 2$, the following inequality holds.*

$$\left\langle \frac{W_{\text{ov}}^{(t)}}{\|W_{\text{ov}}^{(t)}\|}, \frac{W_{\text{ov}}^*}{\|W_{\text{ov}}^*\|} \right\rangle \geq 1 - \frac{12\|W_{\text{ov}}^*\|^3 \log(2|\mathcal{V}|) \log t}{t\eta_0}.$$

*Moreover, $\frac{t\eta_0}{2\|W_{\text{ov}}^*\|} \leq \|W_{\text{ov}}^{(t)}\| \leq t\eta_0$.*

*Proof.* The second argument about the norm of $W_{\text{ov}}^{(t)}$ follows directly from Lemma 1. We aim to prove the first part as follows.

Let $\alpha_t = \frac{2\|W_{\text{ov}}^*\|}{\|W_{\text{ov}}^{(t)}\|} \log(2|\mathcal{V}|)$. By Lemma 2 and the updating rule Equation (7), we have

$$\left\langle W_{\text{ov}}^{(t+1)} - W_{\text{ov}}^{(t)}, \frac{W_{\text{ov}}^*}{\|W_{\text{ov}}^*\|} \right\rangle = -\eta_0 \left\langle \nabla_{W_{\text{ov}}} \mathcal{L}(W_{\text{ov}}^{(t)}), \frac{W_{\text{ov}}^*}{\|W_{\text{ov}}^*\|} \right\rangle$$

$$\geq -\frac{\eta_0}{1 + \alpha_t} \left\langle \nabla_{W_{\text{ov}}} \mathcal{L}(W_{\text{ov}}^{(t)}), \frac{W_{\text{ov}}^{(t)}}{\|W_{\text{ov}}^{(t)}\|} \right\rangle$$

$$= \frac{1}{1 + \alpha_t} \left\langle W_{\text{ov}}^{(t+1)} - W_{\text{ov}}^{(t)}, \frac{W_{\text{ov}}^{(t)}}{\|W_{\text{ov}}^{(t)}\|} \right\rangle$$

$$= \left( 1 - \frac{\alpha_t}{1 + \alpha_t} \right) \left\langle W_{\text{ov}}^{(t+1)} - W_{\text{ov}}^{(t)}, \frac{W_{\text{ov}}^{(t)}}{\|W_{\text{ov}}^{(t)}\|} \right\rangle$$

$$= \frac{1}{2\|W_{\text{ov}}^{(t)}\|} \left( \|W_{\text{ov}}^{(t+1)}\|^2 - \|W_{\text{ov}}^{(t+1)} - W_{\text{ov}}^{(t)}\|^2 - \|W_{\text{ov}}^{(t)}\|^2 \right)$$

$$- \frac{\alpha_t}{1 + \alpha_t} \left\langle W_{\text{ov}}^{(t+1)} - W_{\text{ov}}^{(t)}, \frac{W_{\text{ov}}^{(t)}}{\|W_{\text{ov}}^{(t)}\|} \right\rangle$$

$$\overset{(a)}{=} \frac{\|W_{\text{ov}}^{(t+1)}\|^2 - \|W_{\text{ov}}^{(t)}\|^2}{2\|W_{\text{ov}}^{(t)}\|} - \frac{\eta^2}{2\|W_{\text{ov}}^{(t)}\|}$$

$$+ \frac{\eta_0 \alpha_t}{1 + \alpha_t} \left\langle \frac{\nabla_{W_{\text{ov}}} \mathcal{L}(\theta^{(t)})}{\|\nabla_{W_{\text{ov}}} \mathcal{L}(\theta^{(t)})\|}, \frac{W_{\text{ov}}^{(t)}}{\|W_{\text{ov}}^{(t)}\|} \right\rangle$$

$$\overset{(b)}{\geq} \|W_{\text{ov}}^{(t+1)}\| - \|W_{\text{ov}}^{(t)}\| - \frac{\eta_0^2}{2\|W_{\text{ov}}^{(t)}\|} - \frac{\eta_0 \alpha_t}{1 + \alpha_t},$$

where $(a)$ follows from that $\|W_{\text{ov}}^{(t+1)} - W_{\text{ov}}^{(t)}\| = \eta_0$, and $(b)$ is due to the fact that $x^2 - y^2 \geq 2y(x-y)$ for any $x, y \in \mathbb{R}$.

Summing over $t$ starting from 2, we have

$$\left\langle W_{\text{ov}}^{(t)} - W_{\text{ov}}^{(2)}, \frac{W_{\text{ov}}^*}{\|W_{\text{ov}}^*\|} \right\rangle \geq \|W_{\text{ov}}^{(t)}\| - \|W_{\text{ov}}^{(2)}\| - \sum_{t'=2}^{t-1} \frac{\eta_0^2}{2\|W_{\text{ov}}^{(t')}\|} - \sum_{t'=2}^{t-1} \frac{\eta_0 \alpha_{t'}}{1 + \alpha_{t'}}.$$

Furthermore, due to Lemma 1,

$$\sum_{t'=2}^{t-1} \frac{1}{\|W_{\text{ov}}^{(t')}\|} \leq \sum_{t'=2}^{t-1} \frac{2\|W_{\text{ov}}^*\|/\eta_0}{t}$$

$$\leq \frac{2\|W_{\text{ov}}^*\|}{\eta_0} \log t.$$

Similarly,

$$\sum_{t'=2}^{t-1} \frac{\alpha_{t'}}{1 + \alpha_{t'}} \leq \sum_{t'=2}^{t-1} \frac{2\|W_{\text{ov}}^*\| \log(2|\mathcal{V}|)}{\|W_{\text{ov}}^{(t')}\|}$$

$$\leq \frac{4\|W_{\text{ov}}^*\|^2 \log(2|\mathcal{V}|)}{\eta_0} \log t$$

Therefore,

$$\left\langle \frac{W_{\text{ov}}^{(t)}}{\|W_{\text{ov}}^{(t)}\|}, \frac{W_{\text{ov}}^*}{\|W_{\text{ov}}^*\|} \right\rangle \geq 1 - \frac{\|W_{\text{ov}}^{(2)}\| + 2\eta_0\|W_{\text{ov}}^*\| \log t + 4\|W_{\text{ov}}^*\|^2 \log(2|\mathcal{V}|) \log t}{\|W_{\text{ov}}^{(t)}\|}$$

$$\overset{(a)}{\geq} 1 - \frac{12\|W_{\text{ov}}^*\|^3 \log(2|\mathcal{V}|) \log t}{t\eta_0},$$

where $(a)$ follows from Lemma 1 and $\|W_{\text{ov}}^{(2)}\| \leq 2\eta_0 \leq \|W_{\text{ov}}^*\|$, and $\|W_{\text{ov}}^{(t)}\| \geq t\eta_0/(2\|W_{\text{ov}}^*\|)$

■

## C  Proof of Theorem 1 and Theorem 2

### C.1  Supporting Lemmas

**Lemma 3** *With zero initialization, under the updating rule Equation* (7), *for any iteration $t$, $W_{\text{ov}}^{(t)}$ satisfies that*

$$(e_i - e_{i'})^\top W_{\text{ov}}^{(t)} x = 0, \quad \forall i, i' \neq \text{In}(x).$$

*Proof.* The proof follows directly from induction and the fact that

$$(e_i - e_{i'})^\top W_{\text{ov}}^{(t+1)} x = (e_i - e_{i'})^\top W_{\text{ov}}^{(t)} x - \frac{\eta_0([\text{T}_0^{(t)}]_i - [\text{T}_0^{(t)}]_{i'})}{\|\nabla_{W_{\text{ov}}} \mathcal{L}_0(W_{\text{ov}}^{(t)})\|}, \quad \forall i, i' \neq \text{In}(x).$$

■

**Corollary 1** *Under the settings in Proposition 1, let* $T \geq 384\|W_{\mathrm{ov}}^*\|^5 \log(2|\mathcal{V}|) \log T / \eta_0$, *and* $\Delta = T\eta_0 / (4\|W_{\mathrm{ov}}^*\|^2)$

$$
\begin{cases}
(e_{\mathrm{In}(x)} - e_i)^\top W_{\mathrm{ov}} x \in (\Delta, 3\Delta), \forall i \neq \mathrm{In}(x) \\
(e_i - e_{i'})^\top W_{\mathrm{ov}} x = 0, \forall i, i' \neq \mathrm{In}(x)
\end{cases}
$$

*Proof.* The second equality follows directly from Lemma 3.

To show the first equation, we analyze

$$
(e_{\mathrm{In}(x)} - e_i)^\top W_{\mathrm{ov}}^{(T)} x
$$

$$
= (e_{\mathrm{In}(x)} - e_i)^\top \frac{W_{\mathrm{ov}}^* \|W_{\mathrm{ov}}^{(T)}\|}{\|W_{\mathrm{ov}}^*\|} x + \|W_{\mathrm{ov}}^{(T)}\| (e_{\mathrm{In}(x)} - e_i)^\top \left( \frac{W_{\mathrm{ov}}^{(T)}}{\|W_{\mathrm{ov}}^{(T)}\|} - \frac{W_{\mathrm{ov}}^*}{\|W_{\mathrm{ov}}^*\|} \right) W_{\mathrm{ov}}^{(T)} x
$$

$$
\overset{(a)}{=} \frac{\|W_{\mathrm{ov}}^{(T)}\|}{\|W_{\mathrm{ov}}^*\|} - 2\sqrt{2}\|W_{\mathrm{ov}}^{(T)}\| \sqrt{\frac{12\|W_{\mathrm{ov}}^*\|^3 \log(2|\mathcal{V}|) \log T}{T\eta_0}}
$$

$$
\overset{(b)}{\geq} \frac{T\eta_0}{2\|W_{\mathrm{ov}}^*\|^2} - \sqrt{24 T\eta_0 \|W_{\mathrm{ov}}^*\| \log(2|\mathcal{V}|) \log T}
$$

$$
\geq \frac{T\eta_0}{4\|W_{\mathrm{ov}}^*\|^2},
$$

where $(a)$ follows from Proposition 1, and $(b)$ is due to Lemma 1. On the other hand, we also have

$$
(e_{\mathrm{In}(x)} - e_i)^\top W_{\mathrm{ov}}^{(T)} x \leq \frac{\|W_{\mathrm{ov}}^{(T)}\|}{\|W_{\mathrm{ov}}^*\|} + 2\sqrt{2}\|W_{\mathrm{ov}}^{(T)}\| \sqrt{\frac{12\|W_{\mathrm{ov}}^*\|^3 \log(2|\mathcal{V}|) \log T}{T\eta_0}}
$$

$$
\leq \frac{3T\eta_0}{4\|W_{\mathrm{ov}}^*\|^2}
$$

The proof is finished. ∎

Thus, for simplicity, we further assume that $(e_{\mathrm{In}(x)} - e_i) W_{\mathrm{ov}}^{(T)} x = \Delta$ for all $x$, because $(e_{\mathrm{In}(x)} - e_i) \hat{W}_{\mathrm{ov}} x = \Theta(\Delta)$ for large enough iteration. Next, we provide the general form of the projection of the gradient of Key-Query matrix $W_{\mathrm{kq}}$ follows from a notation for the token weight.

The token weight $\varphi_\ell^{(n,t)}$ of the token $x_\ell^{(n)}$ in the sentence $X^{(n)} = [x_1^{(n)}, \ldots, x_L^{(n)}]$ under $\theta = (W_{\mathrm{ov}}^{(T)}, W_{\mathrm{kq}})$ is calculated as

$$
\varphi_\ell^{(n,t)} = \frac{\exp\left( (x_\ell^{(n)})^\top W_{\mathrm{kq}} X_{-1}^{(n)} \right)}{\sum_{\ell'=1}^{L} \exp\left( (x_{\ell'}^{(n)})^\top W_{\mathrm{kq}} X_{-1}^{(n)} \right)} \tag{9}
$$

**Lemma 4 (Projection of gradient of $W_{\mathrm{kq}}$)** *If $W_{\mathrm{ov}}$ satisfies that*

$$
\begin{cases}
(e_{\mathrm{In}(x)} - e_i)^\top W_{\mathrm{ov}} x = \Delta, \forall i \neq \mathrm{In}(x) \\
(e_i - e_{i'})^\top W_{\mathrm{ov}} x = 0, \forall i, i' \neq \mathrm{In}(x)
\end{cases}
$$

*we have*

$$
\left\langle \nabla_{W_{\mathrm{kq}}} \mathcal{L}(\theta), W_{\mathrm{kq}}' \right\rangle
$$

$$
= \Delta \sum_n \pi^{(n)} ([\mathrm{T}_\theta^{(n)}]_{\mathrm{In}(X^{(n)})} - 1) \sum_{\ell_* \in l(n)} \varphi_{\ell_*}^{(n,\theta)} \left( x_{\ell_*}^{(n)} - \sum_{\ell'} \varphi_{\ell'}^{(n,\theta)} x_{\ell'}^{(n)} \right)^\top W_{\mathrm{kq}}' X_{-1}^{(n)}
$$

$$
+ \Delta \sum_n \pi^{(n)} \sum_{\ell \notin l(n)} [\mathrm{T}_\theta^{(n)}]_{\mathrm{In}(x_\ell^{(n)})} \varphi_\ell^{(n,\theta)} \left( x_\ell^{(n)} - \sum_{\ell'} \varphi_{\ell'}^{(n,\theta)} x_{\ell'}^{(n)} \right)^\top W_{\mathrm{kq}}' X_{-1}^{(n)}.
$$

*Proof.* Recall that $l(n)$ is the set of indices of the optimal tokens in the sample $X^{(n)}$. Thus, for any $\ell_* \in l(n)$, $\mathrm{In}(x_{\ell_*}^{(n)}) = \mathrm{In}(X^{(n)})$. In addition, we denote $\mathrm{T}_\theta(X^{(n)})$ by $\mathrm{T}_\theta^{(n)}$ for simplicity. From Equation (6), we have

$$\left\langle \nabla_{W_{\mathrm{kq}}} \mathcal{L}(\theta), W_{\mathrm{kq}}' \right\rangle$$

$$= \sum_n \pi^{(n)} \sum_\ell (\mathrm{T}_\theta^{(n)} - p^{(n)})^\top W_{\mathrm{ov}} x_\ell^{(n)} \varphi_\ell^{(n,\theta)} \left( x_\ell^{(n)} - \sum_{\ell'} \varphi_{\ell'}^{(n,\theta)} x_{\ell'}^{(n)} \right)^\top W_{\mathrm{kq}}' X_{-1}^{(n)}$$

$$\overset{(a)}{=} \sum_n \pi^{(n)} \sum_\ell \sum_i [\mathrm{T}_\theta^{(n)}]_i (e_i - e_{\mathrm{In}(X^{(n)})})^\top W_{\mathrm{ov}} x_\ell^{(n)} \varphi_\ell^{(n,\theta)} \left( x_\ell^{(n)} - \sum_{\ell'} \varphi_{\ell'}^{(n,\theta)} x_{\ell'}^{(n)} \right)^\top W_{\mathrm{kq}}' X_{-1}^{(n)}$$

$$= \sum_n \pi^{(n)} \sum_{\ell \in l(n)} \sum_i [\mathrm{T}_\theta^{(n)}]_i (e_i - e_{\mathrm{In}(X^{(n)})})^\top W_{\mathrm{ov}} x_\ell^{(n)} \varphi_\ell^{(n,\theta)} \left( x_\ell^{(n)} - \sum_{\ell'} \varphi_{\ell'}^{(n,\theta)} x_{\ell'}^{(n)} \right)^\top W_{\mathrm{kq}}' X_{-1}^{(n)}$$

$$+ \sum_n \pi^{(n)} \sum_{\ell \notin l(n)} \sum_i [\mathrm{T}_\theta^{(n)}]_i (e_i - e_{\mathrm{In}(X^{(n)})})^\top W_{\mathrm{ov}} x_\ell^{(n)} \varphi_\ell^{(n,\theta)} \left( x_\ell^{(n)} - \sum_{\ell'} \varphi_{\ell'}^{(n,\theta)} x_{\ell'}^{(n)} \right)^\top W_{\mathrm{kq}}' X_{-1}^{(n)}$$

$$= \sum_n \pi^{(n)} \sum_{\ell \in l(n)} \sum_{i \neq \mathrm{In}(X^{(n)})} [\mathrm{T}_\theta^{(n)}]_i (-\Delta) \varphi_\ell^{(n,\theta)} \left( x_\ell^{(n)} - \sum_{\ell'} \varphi_{\ell'}^{(n,\theta)} x_{\ell'}^{(n)} \right)^\top W_{\mathrm{kq}}' X_{-1}^{(n)}$$

$$+ \sum_n \pi^{(n)} \sum_{\ell \notin l(n)} [\mathrm{T}_\theta^{(n)}]_{\mathrm{In}(x_\ell^{(n)})} \varphi_\ell^{(n,\theta)} \left( x_\ell^{(n)} - \sum_{\ell'} \varphi_{\ell'}^{(n,\theta)} x_{\ell'}^{(n)} \right)^\top W_{\mathrm{kq}}' X_{-1}^{(n)}$$

$$= \Delta \sum_n \pi^{(n)} ([\mathrm{T}_\theta^{(n)}]_{\mathrm{In}(X^{(n)})} - 1) \sum_{\ell_* \in l(n)} \varphi_{\ell_*}^{(n,\theta)} \left( x_{\ell_*}^{(n)} - \sum_{\ell'} \varphi_{\ell'}^{(n,\theta)} x_{\ell'}^{(n)} \right)^\top W_{\mathrm{kq}}' X_{-1}^{(n)}$$

$$+ \Delta \sum_n \pi^{(n)} \sum_{\ell \notin l(n)} [\mathrm{T}_\theta^{(n)}]_{\mathrm{In}(x_\ell^{(n)})} \varphi_\ell^{(n,\theta)} \left( x_\ell^{(n)} - \sum_{\ell'} \varphi_{\ell'}^{(n,\theta)} x_{\ell'}^{(n)} \right)^\top W_{\mathrm{kq}}' X_{-1}^{(n)},$$

where $(a)$ is due to the fact that $\sum_{i \in [|\mathcal{V}|]} [\mathrm{T}_\theta^{(n)}]_i = 1$ for any $\theta, n$.

∎

### Main Steps

The proof consists of three main steps. First, we show that the optimal token weight has a lower bound. Then, we show that the gradient aligns with the optimal direction. Third, we show that the norm of Key-Query matrix grows linearly. Combining these three steps, we can prove the Theorem 1 and Theorem 2.

Recall that the updating rule for $W_{\mathrm{kq}}^{(t)}$ is

$$W_{\mathrm{kq}}^{(t+1)} = W_{\mathrm{kq}}^{(t)} - \eta \frac{\nabla_{W_{\mathrm{kq}}} \mathcal{L}(\theta^{(t)})}{\|\nabla_{W_{\mathrm{kq}}} \mathcal{L}(\theta^{(t)})\|}. \tag{10}$$

### C.2 Step 1

We first show that the optiaml token weight has a lower bound during the training.

**Lemma 5 (Lower bound of optimal token weight)** *Under the zero initialization and updating rule Equation (10), for any iteration $t$, and any sample $X^{(n)}$, if $l(n)$ is the set of indices of the optimal token in $X^{(n)}$, the following inequality holds.*

$$\varphi_\ell^{(n,t)} \geq \varphi_\ell^{(n,0)} \geq 1/L_{\max}, \quad \forall \ell \in l(n)$$

*Proof.*

First, we introduce the notation that $\varphi_+^{(n,t)} = \sum_{\ell_* \in l(n)} \varphi_{\ell_*}^{(n,t)}$ as the summation of optimal token weights, and $\varphi_-^{(n,t)} = 1 - \varphi_+^{(n,t)}$ as the summation of non-optimal token weights.

At $t = 0$, due to zero initialization, we have $\varphi_\ell^{(n,0)} = 1/L^{(n)}$. Moreover, by Assumption 3, for any $\ell \notin l(n)$, we have $\varphi_+^{(n,0)} \geq q_n(x_\ell)\varphi_\ell^{(n,0)}$.

We perform induction the hypothesis: $\varphi_+^{(n,t)} \geq \varphi_+^{(n,t-1)}$ and $\varphi_{\ell_*}^{(n,t)} \geq \varphi_\ell^{(n,t)}$ for all $\ell_* \in l(n)$ and $\ell \notin l(n)$.

Suppose the hypothesis holds for iteration $t$. Let $x_{\ell_*}^{(n)}$ be the optimal token in the sequence $X^{(n)}$.

Fix a sample $X^{(n')}$. we have

$$(x_{\ell_*}^{(n')})^\top \nabla_{W_{kq}} \mathcal{L}^{(t)}(\theta^{(t)}) X_{-1}^{(n')}$$

$$= \sum_n \pi^{(n)}([T_\theta^{(n)}]_{\mathrm{In}(X^{(n)})} - 1)\varphi_+^{(n,t)} \left( \sum_{\ell' \notin l(n)} \varphi_{\ell'}^{(n,t)}(x_{\ell_*}^{(n)} - x_{\ell'}^{(n)})^\top x_{\ell_*}^{(n')} \right) \left\langle X_{-1}^{(n)}, X_{-1}^{(n')} \right\rangle$$

$$+ \sum_n \pi^{(n)} \sum_{\ell \notin l(n)} [T_\theta^{(n)}]_{\mathrm{In}(x_\ell^{(n)})} \varphi_\ell^{(n,t)} \left( \sum_{\ell'} \varphi_{\ell'}^{(n,t)}(x_\ell^{(n)} - x_{\ell'}^{(n)})^\top x_{\ell_*}^{(n')} \right) \left\langle X_{-1}^{(n)}, X_{-1}^{(n')} \right\rangle$$

Because $\sum_{\ell' \notin l(n)}(x_{\ell'}^{(n)})x_{\ell_*}^{(n')} = 0$ due to Assumption 1, and $(x_{\ell_*}^{(n)})^\top x_{\ell_*}^{(n')} \geq 0$, we immediately have

$$(x_{\ell_*}^{(n')})^\top \nabla_{W_{kq}} \mathcal{L}^{(t)}(\theta^{(t)}) X_{-1}^{(n')} \leq 0.$$

Let $x_{\ell_0}^{(n')}$ be any non-optiaml token in the sequence $X^{(n')}$. Then, we have

$$(x_{\ell_0}^{(n')})^\top \nabla_{W_{kq}} \mathcal{L}^{(t)}(\theta^{(t)}) X_{-1}^{(n')}$$

$$= \sum_n \pi^{(n)}([T_\theta^{(n)}]_{\mathrm{In}(X^{(n)})} - 1)\varphi_+^{(n,t)} \left( \sum_{\ell' \notin l(n)} \varphi_{\ell'}^{(n,t)}(x_{\ell_*}^{(n)} - x_{\ell'}^{(n)})^\top x_{\ell_0}^{(n')} \right) \left\langle X_{-1}^{(n)}, X_{-1}^{(n')} \right\rangle$$

$$+ \sum_n \pi^{(n)} \sum_{\ell \notin l(n)} [T_\theta^{(n)}]_{\mathrm{In}(x_\ell^{(n)})} \varphi_\ell^{(n,t)} \left( \sum_{\ell'} \varphi_{\ell'}^{(n,t)}(x_\ell^{(n)} - x_{\ell'}^{(n)})^\top x_{\ell_0}^{(n')} \right) \left\langle X_{-1}^{(n)}, X_{-1}^{(n')} \right\rangle$$

$$= \sum_n \pi^{(n)}([T_\theta^{(n)}]_{\mathrm{In}(X^{(n)})} - 1)\varphi_+^{(n,t)} \left( \sum_{\ell' \notin l(n)} \varphi_{\ell'}^{(n,t)}(-x_{\ell'}^{(n)})^\top x_{\ell_0}^{(n')} \right) \left\langle X_{-1}^{(n)}, X_{-1}^{(n')} \right\rangle$$

$$+ \sum_n \pi^{(n)} \sum_{\ell \notin l(n)} [T_\theta^{(n)}]_{\mathrm{In}(x_\ell^{(n)})} \varphi_\ell^{(n,t)} \left( \sum_{\ell' \notin l(n)} \varphi_{\ell'}^{(n,t)}(x_\ell^{(n)} - x_{\ell'}^{(n)})^\top x_{\ell_0}^{(n')} \right) \left\langle X_{-1}^{(n)}, X_{-1}^{(n')} \right\rangle$$

$$\geq \sum_n \pi^{(n)} \left( (1 - [T_\theta^{(n)}]_{\mathrm{In}(X^{(n)})})\varphi_{\ell_+}^{(n,t)} - \sum_{\ell \notin l(n)} [T_\theta^{(n)}]_{\mathrm{In}(x_\ell^{(n)})} \varphi_\ell^{(n,t)} \right) \left( \sum_{\ell' \notin l(n)} \varphi_{\ell'}^{(n,t)}(x_{\ell'}^{(n)})^\top x_{\ell_0}^{(n')} \right) \left\langle X_{-1}^{(n)}, X_{-1}^{(n')} \right\rangle$$

$$\geq 0,$$

where the last inequality is due to Assumption 3, the induction hypothesis and $\sum_{i \in [|\mathcal{V}|]}[T_\theta^{(n)}]_i = 1$

Therefore, for any $n$, we have

$$\varphi_{\ell_*}^{(n,t+1)} = \frac{\exp\left( (x_{\ell_*}^{(n)})^\top W_{kq}^{(t+1)} X_{-1}^{(n)} \right)}{\sum_\ell \exp\left( (x_\ell^{(n)})^\top W_{kq}^{(t+1)} X_{-1}^{(n)} \right)}$$

$$= \frac{\exp\left((x_{\ell_*}^{(n)})^\top W_{\mathrm{kq}}^{(t)} X_{-1}^{(n)} - \eta(x_{\ell_*}^{(n)})^\top \nabla_{W_{\mathrm{kq}}} \mathcal{L}^{(t)}(\theta^{(t)}) X_{-1}^{(n)} / \|\nabla_{W_{\mathrm{kq}}} \mathcal{L}^{(t)}(\theta^{(t)})\|\right)}{\sum_\ell \exp\left((x_\ell^{(n)})^\top W_{\mathrm{kq}}^{(t)} X_{-1}^{(n)} - \eta(x_\ell^{(n)})^\top \nabla_{W_{\mathrm{kq}}} \mathcal{L}^{(t)}(\theta^{(t)}) X_{-1}^{(n)} / \|\nabla_{W_{\mathrm{kq}}} \mathcal{L}^{(t)}(\theta^{(t)})\|\right)}$$

$$= \frac{\exp\left((x_{\ell_*}^{(n)})^\top W_{\mathrm{kq}}^{(t)} X_{-1}^{(n)}\right)}{\sum_\ell \exp\left((x_\ell^{(n)})^\top W_{\mathrm{kq}}^{(t)} X_{-1}^{(n)} + \eta(x_{\ell_*}^{(n)} - x_\ell^{(n)})^\top \nabla_{W_{\mathrm{kq}}} \mathcal{L}^{(t)}(\theta^{(t)}) X_{-1}^{(n)} / \|\nabla_{W_{\mathrm{kq}}} \mathcal{L}^{(t)}(\theta^{(t)})\|\right)}$$

$$\geq \frac{\exp\left((x_{\ell_*}^{(n)})^\top W_{\mathrm{kq}}^{(t)} X_{-1}^{(n)}\right)}{\sum_\ell \exp\left((x_\ell^{(n)})^\top W_{\mathrm{kq}}^{(t)} X_{-1}^{(n)}\right)}$$

$$= \varphi_{\ell_*}^{(n,t)},$$

which implies that $\varphi_+^{(n,t+1)} \geq \varphi_+^{(n,t)}$.

For the second argument in the hypothesis, we examine $\varphi_{\ell_*}^{(n,t+1)} / \varphi_\ell^{(n,t+1)}$ for any $\ell \notin l(n)$. We have

$$\frac{\varphi_{\ell_*}^{(n,t+1)}}{\varphi_\ell^{(n,t+1)}} = \exp\left((x_{\ell_*}^{(n)} - x_\ell^{(n)})^\top W_{\mathrm{kq}}^{(t+1)} X_{-1}^{(n)}\right)$$

$$= \exp\left((x_{\ell_*}^{(n)} - x_\ell^{(n)})^\top W_{\mathrm{kq}}^{(t)} X_{-1}^{(n)}\right) \exp\left(-\frac{\eta}{\|\nabla_{W_{\mathrm{kq}}} \mathcal{L}^{(t)}(\theta^{(t)})\|} (x_{\ell_*}^{(n)} - x_\ell^{(n)})^\top \nabla_{W_{\mathrm{kq}}} \mathcal{L}^{(t)}(\theta^{(t)}) X_{-1}^{(n)}\right)$$

$$\geq \frac{\varphi_{\ell_*}^{(n,t)}}{\varphi_\ell^{(n,t)}}$$

$$\geq 1.$$

The proof is finished.

∎

## C.3  Step 2

The following lemma shows that the norm of the Key-Query Matrix increases linearly with the number of iterations.

**Lemma 6** *Under the initialization $W_{\mathrm{kq}}^{(0)}$ and the updating rule Equation (10), for each iteration $t$, the following inequality holds.*

$$t\eta + \|W_{\mathrm{kq}}^{(0)}\| \geq \|W_{\mathrm{kq}}^{(t)}\| \geq \frac{t\eta}{2 L_{\max} \|W_{\mathrm{kq}}^*\|} - \|W_{\mathrm{kq}}^{(0)}\|.$$

*Proof.*

We examine the gradient $\nabla_{W_{\mathrm{kq}}} \mathcal{L}(\theta)$ projected onto the optimal direction $W_{\mathrm{kq}}^* / \|W_{\mathrm{kq}}^*\|$.

$$\left\langle \nabla_{W_{\mathrm{kq}}} \mathcal{L}(\theta^{(t)}), W_{\mathrm{kq}}^* \right\rangle$$

$$= \sum_n \pi^{(n)} \Delta \sum_{\ell_* \in l(n)} ([\mathrm{T}_{\theta^{(t)}}^{(n)}]_{\mathrm{In}(x_{\ell_*}^{(n)})} - 1) \varphi_{\ell_*}^{(n,t)} \left( a_{\ell_*}^{(n,*)} - \sum_{\ell'} \varphi_{\ell'}^{(n,t)} a_{\ell'}^{(n,*)} \right)$$

$$+ \sum_n \pi^{(n)} \Delta \sum_{\ell \notin l(n)} [\mathrm{T}_{\theta^{(t)}}^{(n)}]_{\mathrm{In}(x_\ell^{(n)})} \varphi_\ell^{(n,t)} \left( a_\ell^{(n,*)} - \sum_{\ell'} \varphi_{\ell'}^{(n,t)} a_{\ell'}^{(n,*)} \right)$$

$$= \sum_n \pi^{(n)} \Delta ([\mathrm{T}_{\theta^{(t)}}^{(n)}]_{\mathrm{In}(X^{(n)})} - 1) \varphi_+^{(n,t)} \left( \sum_{\ell' \notin l(n)} \varphi_{\ell'}^{(n,t)} (a_{\ell_*}^{(n,*)} - a_{\ell'}^{(n,*)}) \right)$$

$$+ \sum_n \pi^{(n)} \Delta \sum_{\ell \notin l(n)} [\mathrm{T}_{\theta^{(t)}}^{(n)}]_{\mathrm{In}(x_\ell^{(n)})} \varphi_\ell^{(n,t)} \left( \sum_{\ell' \in l(n)} \varphi_{\ell'}^{(n,t)} (a_\ell^{(n,*)} - a_{\ell'}^{(n,*)}) \right)$$

$$\leq \sum_n \pi^{(n)} \Delta ([\mathrm{T}_{\theta^{(t)}}^{(n)}]_{\mathrm{In}(X^{(n)})} - 1) \varphi_+^{(n,t)} \varphi_-^{(n,t)}$$

$$+ \sum_n \pi^{(n)} \Delta \sum_{\ell \notin l(n)} [\mathrm{T}_{\theta^{(t)}}^{(n)}]_{\mathrm{In}(x_\ell^{(n)})} \varphi_\ell^{(n,t)} (-\varphi_+^{(n,t)}),$$

where $\varphi_+^{(n,t)} = \sum_{\ell_* \in l(n)} \varphi_{\ell_*}^{(n,t)}$ is the summation of optimal token weights, and $\varphi_-^{(n,t)} = 1 - \varphi_+^{(n,t)}$ is the summation of non-optimal token weights.

On the other hand,

$$\|\nabla_{W_{\mathrm{kq}}} \mathcal{L}(\theta^{(t)})\|$$

$$= \left\langle \nabla_{W_{\mathrm{kq}}} \mathcal{L}(\theta^{(t)}), \frac{\nabla_{W_{\mathrm{kq}}} \mathcal{L}(\theta^{(t)})}{\|\nabla_{W_{\mathrm{kq}}} \mathcal{L}(\theta^{(t)})\|} \right\rangle$$

$$= \sum_n \pi^{(n)} \Delta \sum_{\ell_* \in l(n)} ([\mathrm{T}_{\theta^{(t)}}^{(n)}]_{\mathrm{In}(x_{\ell_*}^{(n)})} - 1) \varphi_{\ell_*}^{(n,t)} \left( x_{\ell_*}^{(n)} - \sum_{\ell'} \varphi_{\ell'}^{(n,t)} x_{\ell'}^{(n)} \right)^\top \frac{\nabla_{W_{\mathrm{kq}}} \mathcal{L}(\theta^{(t)})}{\|\nabla_{W_{\mathrm{kq}}} \mathcal{L}(\theta^{(t)})\|} X_{-1}^{(n)}$$

$$+ \sum_n \pi^{(n)} \Delta \sum_{\ell \notin l(n)} [\mathrm{T}_{\theta^{(t)}}^{(n)}]_{\mathrm{In}(x_\ell^{(n)})} \varphi_\ell^{(n,t)} \left( x_\ell^{(n)} - \sum_{\ell'} \varphi_{\ell'}^{(n,t)} x_{\ell'}^{(n)} \right)^\top \frac{\nabla_{W_{\mathrm{kq}}} \mathcal{L}(\theta^{(t)})}{\|\nabla_{W_{\mathrm{kq}}} \mathcal{L}(\theta^{(t)})\|} X_{-1}^{(n)}$$

$$\leq 2 \sum_n \pi^{(n)} \Delta (1 - [\mathrm{T}_\theta^{(n)}]_{\mathrm{In}(X^{(n)})}) \varphi_+^{(n,t)} \varphi_-^{(n,t)} + 2 \sum_n \pi^{(n)} \Delta \sum_{\ell \notin l(n)} [\mathrm{T}_\theta^{(n)}]_{\mathrm{In}(x_\ell^{(n)})} \varphi_\ell^{(n,t)}$$

Thus, we have

$$\left\langle \frac{\nabla_{W_{\mathrm{kq}}} \mathcal{L}(\theta)}{\|\nabla_{W_{\mathrm{kq}}} \mathcal{L}(\theta)\|}, \frac{W_{\mathrm{kq}}^*}{\|W_{\mathrm{kq}}^*\|} \right\rangle \leq - \frac{\min_n \varphi_+^{(n,t)}}{2\|W_{\mathrm{kq}}^*\|} \leq - \frac{1}{2 L_{\max} \|W_{\mathrm{kq}}^*\|}$$

By the updating rule Equation (10), we have

$$\|W_{\mathrm{kq}}^{(t)}\| = \left\| W_{\mathrm{kq}}^{(0)} - \sum_{t'=0}^{t-1} \eta \frac{\nabla_{W_{\mathrm{kq}}} \mathcal{L}(\theta^{(t')})}{\|\nabla_{W_{\mathrm{kq}}} \mathcal{L}(\theta^{(t')})\|} \right\|$$

$$\geq \left\langle W_{\mathrm{kq}}^{(0)}, \frac{W_{\mathrm{kq}}^*}{\|W_{\mathrm{kq}}^*\|} \right\rangle - \sum_{t' \leq t-1} \eta \left\langle \frac{\nabla_{W_{\mathrm{kq}}} \mathcal{L}(\theta^{(t')})}{\|\nabla_{W_{\mathrm{kq}}} \mathcal{L}(\theta^{(t')})\|}, \frac{W_{\mathrm{kq}}^*}{\|W_{\mathrm{kq}}^*\|} \right\rangle$$

$$\geq \sum_{t' < t} \frac{\eta}{2 L_{\max} \|W_{\mathrm{kq}}^*\|} - \|W_{\mathrm{kq}}^{(0)}\|$$

$$= \frac{t\eta}{2 L_{\max} \|W_{\mathrm{kq}}^*\|} - \|W_{\mathrm{kq}}^{(0)}\|.$$

In addition, by the triangle inequality,

$$\|W_{\mathrm{kq}}^{(t)}\| = \left\| W_{\mathrm{kq}}^{(0)} - \sum_{t'=0}^{t-1} \eta \frac{\nabla_{W_{\mathrm{kq}}} \mathcal{L}(\theta^{(t')})}{\|\nabla_{W_{\mathrm{kq}}} \mathcal{L}(\theta^{(t')})\|} \right\|$$

$$\leq t\eta + \|W_{\mathrm{kq}}^{(0)}\|.$$

The proof is completed. ∎

## C.4 Step 3

We next show that the gradient $\nabla_{W_{\text{kq}}} \mathcal{L}(\theta^{(t)})$ is close to the optimal direction $W_{\text{kq}}^*$.

**Lemma 7 (Gradient aligns with the optimal direction)** *Let* $t_0 = \lceil \frac{8L_{\max}\|W_{\text{kq}}^*\|^2}{\eta} \rceil$. *Then, for any* $t \geq t_0$, *we have*

$$\left\langle \nabla_{W_{\text{kq}}} \mathcal{L}(\theta^{(t)}), W_{\text{kq}}^{(t)} \right\rangle \geq (1 + \alpha_t) \left\langle \nabla_{W_{\text{kq}}} \mathcal{L}(\theta^{(t)}), W_{\text{kq}}^* \right\rangle \frac{\|W_{\text{kq}}^{(t)}\|}{\|W_{\text{kq}}^*\|}$$

*where*

$$\alpha_t = \frac{4N L_{\max}^2 \|W_{\text{kq}}^*\|^2}{\|W_{\text{kq}}^{(t)}\|} \left( 1 + \log \left( 2L_{\max}\|W_{\text{kq}}^{(t)}\| \right) \right)$$

*Proof.* During the proof, we denote

$$\begin{cases} a_\ell^{(n,t)} = (x_\ell^{(n)})^\top W_{\text{kq}}^{(t)} X_{-1}^{(n)} \\ a_\ell^{(n,*)} = (x_\ell^{(n)})^\top W_{\text{kq}}^* X_{-1}^{(n)} \frac{\|W_{\text{kq}}^{(t)}\|}{\|W_{\text{kq}}^*\|} \end{cases}$$

$$\beta_0 = \frac{2L_{\max}^2 \|W_{\text{kq}}^*\|^2}{\|W_{\text{kq}}^{(t)}\|} (1 + \log(2L_{\max}\|W_{\text{kq}}^{(t)}\|)).$$

We point out a few facts that will be frequently used in the proof.

If $a_\ell^{(n,t)} \leq a_{\ell'}^{(n,t)} - C_0$, then we have

$$\varphi_\ell^{(n,t)} = \varphi_{\ell'}^{(n,t)} \exp \left( a_\ell^{(n,t)} - a_{\ell'}^{(n,t)} \right) \leq \exp(-C_0) \tag{11}$$

The same result holds if $a_{\ell'}^{(n,t)}$ is replaced any convex combination of a set of $a_{\ell'}^{(n,t)}$'s.

We start the proof by noting that $W_{\text{kq}}^*$ is the minimum unique solution to the problem

$$W_{\text{kq}}^* = \arg\min \|W\|, \quad \text{s.t.} \quad (x_{\ell_*}^{(n)} - x_\ell^{(n)})W X_{-1}^{(n)} \geq 1, \quad \forall \ell_* \in l^{(n)}, \ell \notin l^{(n)}, \forall n. \tag{12}$$

Therefore, if $W_{\text{kq}}^{(t)} \frac{\|W_{\text{kq}}^*\|}{\|W_{\text{kq}}^{(t)}\|} = W_{\text{kq}}^*$, the results is trivial since $\left\langle \nabla_{W_{\text{kq}}} \mathcal{L}(\theta^{(t)}), W_{\text{kq}}^* \right\rangle \leq 0$.

In the following, we focus on the case when $W_{\text{kq}}^{(t)} \frac{\|W_{\text{kq}}^*\|}{\|W_{\text{kq}}^{(t)}\|} \neq W_{\text{kq}}^*$. Then, there must be at least a sentence $X^{(n)}$, such that $W_{\text{kq}}^{(t)} \frac{\|W_{\text{kq}}^*\|}{\|W_{\text{kq}}^{(t)}\|}$ violates the contraint on $X^{(n)}$. In other words, we must have

$$a_{\ell_*}^{(n,t)} - a_\ell^{(n,t)} = (x_{\ell_*}^{(n)} - x_\ell^{(n)})W_{\text{kq}}^{(t)} X_{-1}^{(n)} \leq \frac{\|W_{\text{kq}}^{(t)}\|}{\|W_{\text{kq}}^*\|}.$$

This implies that for those $n$, we must have $\varphi_\ell^{(n,t)} \geq \exp(-\frac{\|W_{\text{kq}}^{(t)}\|}{\|W_{\text{kq}}^*\|})$

Thus, we consider two types of samples in the folloiwng.

**Type 1.** Let us consider $X^{(n)}$ such that $\varphi_-^{(n,t)} \geq \exp(-(1 + \beta_0/2)\|W_{\text{kq}}^{(t)}\|/\|W_{\text{kq}}^*\|)$.

Recall that the inner product between the gradient $\nabla_{W_{\text{kq}}} \mathcal{L}(\theta^{(t)})$ and any other Key-Query matrix $\theta' = W_{\text{kq}}'$ has the following form (Lemma 4).

$$\left\langle \nabla_{W_{\text{kq}}} \mathcal{L}(\theta^{(t)}), W_{\text{kq}}' \right\rangle$$

$$= \Delta \sum_n \pi^{(n)} ([\mathrm{T}_\theta^{(n)}]_{\mathrm{In}(X^{(n)})} - 1) \sum_{\ell_* \in l(n)} \varphi_{\ell_*}^{(n,\theta)} \left( x_{\ell_*}^{(n)} - \sum_{\ell'} \varphi_{\ell'}^{(n,\theta)} x_{\ell'}^{(n)} \right)^\top W'_{\mathrm{kq}} X_{-1}^{(n)}$$

$$+ \Delta \sum_n \pi^{(n)} \sum_{\ell \notin l(n)} [\mathrm{T}_\theta^{(n)}]_{\mathrm{In}(x_\ell^{(n)})} \varphi_\ell^{(n,\theta)} \left( x_\ell^{(n)} - \sum_{\ell'} \varphi_{\ell'}^{(n,\theta)} x_{\ell'}^{(n)} \right)^\top W'_{\mathrm{kq}} X_{-1}^{(n)}$$

Let $L_n(\theta) = -\log e_{\mathrm{In}(X^{(n)})}^\top \mathrm{T}_\theta(X^{(n)})$ be the loss on sample $X^{(n)}$.

To proceed, we examine the gradient on each sample $X^{(n)}$ with $\varphi_-^{(n,t)} \geq \exp(-(1 + \beta_0/2)\|W_{\mathrm{kq}}^{(t)}\|/\|W_{\mathrm{kq}}^*\|)$, which can be divided into two parts. $\langle \nabla_{W_{\mathrm{kq}}} L_n(\theta^{(t)}), W_{\mathrm{kq}} \rangle = \Delta(A^{(n,t)} + B^{(n,t)})$, where

$$\begin{cases} A^{(n,t)} = \displaystyle\sum_{\ell_* \in l(n)} ([\mathrm{T}_{\theta^{(t)}}^{(n)}]_{\mathrm{In}(x_{\ell_*}^{(n)})} - 1) \varphi_{\ell_*}^{(n,t)} \left( a_{\ell_*}^{(n,t)} - \sum_{\ell'} \varphi_{\ell'}^{(n,t)} a_{\ell'}^{(n,t)} \right), \\[3mm] B^{(n,t)} = \displaystyle\sum_{\ell \notin l(n)} [\mathrm{T}_{\theta^{(t)}}^{(n)}]_{\mathrm{In}(x_\ell^{(n)})} \varphi_\ell^{(n,t)} \left( a_\ell^{(n,t)} - \sum_{\ell'} \varphi_{\ell'}^{(n,t)} a_{\ell'}^{(n,t)} \right). \end{cases}$$

We further let

$$A^{(n,*)} = \sum_{\ell_* \in l(n)} ([\mathrm{T}_{\theta^{(t)}}^{(n)}]_{\mathrm{In}(x_\ell^{(n)})} - 1) \varphi_{\ell_*}^{(n,t)} \left( a_{\ell_*}^{(n,*)} - \sum_{\ell'} \varphi_{\ell'}^{(n,t)} a_{\ell'}^{(n,*)} \right),$$

and

$$B^{(n,*)} = \sum_{\ell \notin l(n)} [\mathrm{T}_{\theta^{(t)}}^{(n)}]_{\mathrm{In}(x_\ell^{(n)})} \varphi_\ell^{(n,t)} \left( a_\ell^{(n,*)} - \sum_{\ell'} \varphi_{\ell'}^{(n,t)} a_{\ell'}^{(n,*)} \right).$$

Thus, we aim to find the relationship $A^{(n,t)} + B^{(n,t)}$ between $A^{(n,*)} + B^{(n,*)}$.

We first provide the upper bounds for $A^{(n,*)}$ and $B^{(n,*)}$.

$$A^{(n,*)} = \sum_{\ell_* \in l(n)} ([\mathrm{T}_{\theta^{(t)}}^{(n)}]_{\mathrm{In}(x_{\ell_*}^{(n)})} - 1) \varphi_{\ell_*}^{(n,t)} \left( a_{\ell_*}^{(n,*)} - \sum_{\ell'} \varphi_{\ell'}^{(n,t)} a_{\ell'}^{(n,*)} \right)$$

$$= \sum_{\ell_* \in l(n)} ([\mathrm{T}_{\theta^{(t)}}^{(n)}]_{\mathrm{In}(x_{\ell_*}^{(n)})} - 1) \varphi_{\ell_*}^{(n,t)} \left( \sum_{\ell' \notin l(n)} \varphi_{\ell'}^{(n,t)} (a_{\ell_*}^{(n,*)} - a_{\ell'}^{(n,*)}) \right)$$

$$\overset{(a)}{\leq} ([\mathrm{T}_{\theta^{(t)}}^{(n)}]_{\mathrm{In}(X^{(n)})} - 1) \varphi_+^{(n,t)} \varphi_-^{(n,t)} \frac{\|W_{\mathrm{kq}}^{(t)}\|}{\|W_{\mathrm{kq}}^*\|},$$

where $(a)$ is due to the fact that $(x_{\ell_*}^{(n)} - x_\ell^{(n)}) W_{\mathrm{kq}}^* X_{-1}^{(n)} \geq 1$, and $a_\ell^{(n,*)} = (x_\ell^{(n)})^\top W_{\mathrm{kq}}^* X_{-1}^{(n)} \|W_{\mathrm{kq}}^{(t)}\|/\|W_{\mathrm{kq}}^*\|$.

On the other hand

$$A^{(n,t)} = \sum_{\ell \in l(n)} ([\mathrm{T}_{\theta^{(t)}}^{(n)}]_{\mathrm{In}(x_\ell^{(n)})} - 1) \varphi_\ell^{(n,t)} \left( a_\ell^{(n,t)} - \sum_{\ell'} \varphi_{\ell'}^{(n,t)} a_{\ell'}^{(n,t)} \right)$$

$$= \sum_{\ell \in l(n)} ([\mathrm{T}_{\theta^{(t)}}^{(n)}]_{\mathrm{In}(x_\ell^{(n)})} - 1) \varphi_\ell^{(n,t)} \left( \sum_{\ell' \notin l(n)} \varphi_{\ell'}^{(n,t)} (a_\ell^{(n,t)} - a_{\ell'}^{(n,t)}) \right)$$

$$
= \max_{\mathtt{T}} \left\{ \sum_{\ell \in l(n)} ([\mathtt{T}^{(n)}_{\theta^{(t)}}]_{\mathrm{In}(x^{(n)}_\ell)} - 1)\varphi^{(n,t)}_\ell \left( \sum_{\substack{\ell' \notin l(n) \\ \mathrm{diff} < \mathtt{T}}} \varphi^{(n,t)}_{\ell'} \underbrace{(a^{(n,t)}_\ell - a^{(n,t)}_{\ell'})}_{\mathrm{diff}} \right) \right.
$$

$$
\left. + \sum_{\ell \in l(n)} ([\mathtt{T}^{(n)}_{\theta^{(t)}}]_{\mathrm{In}(x^{(n)}_\ell)} - 1)\varphi^{(n,t)}_\ell \left( \sum_{\substack{\ell' \notin l(n) \\ \mathrm{diff} > \mathtt{T}}} \varphi^{(n,t)}_{\ell'} \underbrace{(a^{(n,t)}_\ell - a^{(n,t)}_{\ell'})}_{\mathrm{diff}} \right) \right\}.
$$

$$
\overset{(a)}{\geq} \max_{\mathtt{T}} \left\{ \sum_{\ell \in l(n)} ([\mathtt{T}^{(n)}_{\theta^{(t)}}]_{\mathrm{In}(x^{(n)}_\ell)} - 1)\varphi^{(n,t)}_\ell \left( \sum_{\ell' \notin l(n)} \varphi^{(n,t)}_{\ell'} \mathtt{T} \right) \right.
$$

$$
\left. + \sum_{\ell \in l(n)} ([\mathtt{T}^{(n)}_{\theta^{(t)}}]_{\mathrm{In}(x^{(n)}_\ell)} - 1)\varphi^{(n,t)}_\ell \left( 2 \sum_{\ell' \notin l(n)} \exp(-\mathtt{T}) \|W^{(t)}_{\mathrm{kq}}\| \right) \right\}
$$

$$
\geq \max_{\mathtt{T}} \left\{ \sum_{\ell \in l(n)} ([\mathtt{T}^{(n)}_{\theta^{(t)}}]_{\mathrm{In}(x^{(n)}_\ell)} - 1)\varphi^{(n,t)}_\ell \left( \varphi^{(n,t)}_- \mathtt{T} + 2L_{\max}\exp(-\mathtt{T})\|W^{(t)}_{\mathrm{kq}}\| \right) \right\}
$$

$$
\overset{(b)}{\geq} \sum_{\ell \in l(n)} ([\mathtt{T}^{(n)}_{\theta^{(t)}}]_{\mathrm{In}(x^{(n)}_\ell)} - 1)\varphi^{(n,t)}_\ell \varphi^{(n,t)}_- \left( 1 + \log \frac{2L_{\max}\|W^{(t)}_{\mathrm{kq}}\|}{\varphi^{(n,t)}_-} \right),
$$

where $(a)$ is due to Equation (11), and $(b)$ is obtained by choosing $\mathtt{T} = \log \frac{2L_{\max}\|W^{(t)}_{\mathrm{kq}}\|}{\varphi^{(n,t)}_-}$.

Recall that $\varphi^{(n,t)}_- \geq \exp\left(-(1+\beta_0/2)\|W^{(t)}_{\mathrm{kq}}\|/\|W^*_{\mathrm{kq}}\|\right)$ and $\beta_0 \geq \frac{2\|W^*_{\mathrm{kq}}\|(1+\log(2L_{\max}\|W^{(t)}_{\mathrm{kq}}\|))}{\|W^{(t)}_{\mathrm{kq}}\|}$.
Thus, we further have

$$
A^{(n,t)} \geq ([\mathtt{T}^{(n)}_{\theta^{(t)}}]_{\mathrm{In}(X^{(n)})} - 1)\varphi^{(n,t)}_+ \varphi^{(n,t)}_- \left( 1 + \log \frac{2L_{\max}\|W^{(t)}_{\mathrm{kq}}\|}{\varphi^{(n,t)}_-} \right)
$$

$$
\geq ([\mathtt{T}^{(n)}_{\theta^{(t)}}]_{\mathrm{In}(X^{(n)})} - 1)\varphi^{(n,t)}_+ \varphi^{(n,t)}_- \left( 1 + \log(2L_{\max}\|W^{(t)}_{\mathrm{kq}}\|) + (1+\beta_0/2)\frac{\|W^{(t)}_{\mathrm{kq}}\|}{\|W^*_{\mathrm{kq}}\|} \right)
$$

$$
\geq ([\mathtt{T}^{(n)}_{\theta^{(t)}}]_{\mathrm{In}(X^{(n)})} - 1)\varphi^{(n,t)}_+ \varphi^{(n,t)}_- \left( \frac{\beta_0\|W^{(t)}_{\mathrm{kq}}\|}{2\|W^*_{\mathrm{kq}}\|} + (1+\beta_0/2)\frac{\|W^{(t)}_{\mathrm{kq}}\|}{\|W^*_{\mathrm{kq}}\|} \right)
$$

$$
= (1+\beta_0)([\mathtt{T}^{(n)}_{\theta^{(t)}}]_{\mathrm{In}(X^{(n)})} - 1)\varphi^{(n,t)}_+ \varphi^{(n,t)}_- \frac{\|W^{(t)}_{\mathrm{kq}}\|}{\|W^*_{\mathrm{kq}}\|}
$$

$$
\geq (1+\beta_0)A^{(n,*)}
$$

Next, we analyze $B^{(n,t)}$, and further divide $B^{(n,\theta)}$ into $B^{(n,\theta)}_+$ and $B^{(n,\theta)}_-$:

$$
\begin{cases}
B^{(n,\theta)}_+ = \sum_{\ell \notin l(n)} [\mathtt{T}^{(n)}_{\theta^{(t)}}]_{\mathrm{In}(x^{(n)}_\ell)}\varphi^{(n,t)}_\ell \left( \varphi^{(n,t)}_+ a^{(n,\theta)}_\ell - \sum_{\ell' \in l(n)} \varphi^{(n,t)}_{\ell'} a^{(n,\theta)}_{\ell'} \right) \\[2em]
B^{(n,\theta)}_- = \sum_{\ell \notin l(n)} [\mathtt{T}^{(n)}_{\theta^{(t)}}]_{\mathrm{In}(x^{(n)}_\ell)}\varphi^{(n,t)}_\ell \left( \varphi^{(n,t)}_- a^{(n,\theta)}_\ell - \sum_{\ell' \notin l(n)} \varphi^{(n,t)}_{\ell'} a^{(n,\theta)}_{\ell'} \right)
\end{cases}
$$

Due to Proposition 4, we have $B^{(n,*)}_- = 0$, and thus

$$
B^{(n,*)} = B^{(n,*)}_+ \leq \sum_{\ell \notin l(n)} [\mathtt{T}^{(n)}_{\theta^{(t)}}]_{\mathrm{In}(x^{(n)}_\ell)}\varphi^{(n,t)}_\ell (-\varphi^{(n,t)}_+)\frac{\|W^{(t)}_{\mathrm{kq}}\|}{\|W^*_{\mathrm{kq}}\|} \leq 0.
$$

We then analyze:

$$
\begin{aligned}
&B_+^{(n,t)} - (1+\beta_0)B_+^{(n,*)} \\
&= \sum_{\ell \notin l(n)} [\mathrm{T}_{\theta^{(t)}}^{(n)}]_{\mathrm{In}(x_\ell^{(n)})} \varphi_\ell^{(n,t)} \left( \varphi_+^{(n,t)} a_\ell^{(n,t)} - \varphi_+^{(n,t)} a_{\ell_*}^{(n,t)} - (1+\beta_0)\left( \varphi_+^{(n,t)} a_\ell^{(n,*)} - \varphi_+^{(n,t)} a_{\ell_*}^{(n,*)} \right) \right) \\
&= \sum_{\ell \notin l(n)} [\mathrm{T}_{\theta^{(t)}}^{(n)}]_{\mathrm{In}(x_\ell^{(n)})} \varphi_\ell^{(n,t)} \varphi_+^{(n,t)} \left( \underbrace{a_\ell^{(n,t)} - a_{\ell_*}^{(n,t)}}_{b_{t,\ell}} - \underbrace{(1+\beta_0)\left( a_\ell^{(n,*)} - a_{\ell_*}^{(n,*)} \right)}_{b_{*,\ell}} \right) \\
&\geq \sum_{\substack{\ell \notin l(n) \\ b_{t,\ell} < b_{*,\ell}}} [\mathrm{T}_{\theta^{(t)}}^{(n)}]_{\mathrm{In}(x_\ell^{(n)})} \varphi_\ell^{(n,t)} \varphi_+^{(n,t)} (b_{t,\ell} - b_{*,\ell}) \\
&\overset{(a)}{\geq} \sum_{\ell \notin l(n)} [\mathrm{T}_{\theta^{(t)}}^{(n)}]_{\mathrm{In}(x_\ell^{(n)})} \varphi_+^{(n,t)} \exp\left( -(1+\beta_0)\frac{\|W_{\mathrm{kq}}^{(t)}\|}{\|W_{\mathrm{kq}}^*\|} \right) \left( -2\|W_{\mathrm{kq}}^{(t)}\| \right) \\
&= -2 \sum_{\ell \notin l(n)} [\mathrm{T}_{\theta^{(t)}}^{(n)}]_{\mathrm{In}(x_\ell^{(n)})} \varphi_+^{(n,t)} \exp\left( -(1+\beta_0/2)\frac{\|W_{\mathrm{kq}}^{(t)}\|}{\|W_{\mathrm{kq}}^*\|} \right) \|W_{\mathrm{kq}}^{(t)}\| \exp\left( -\frac{\beta_0}{2}\frac{\|W_{\mathrm{kq}}^{(t)}\|}{\|W_{\mathrm{kq}}^*\|} \right) \\
&\overset{(b)}{\geq} -2L_{\max}(1 - [\mathrm{T}_{\theta^{(t)}}^{(n)}]_{\mathrm{In}(X^{(n)})}) \varphi_+^{(n,t)} \varphi_-^{(n,t)} \frac{\|W_{\mathrm{kq}}^{(t)}\|}{\|W_{\mathrm{kq}}^*\|} \exp\left( -\frac{\beta_0}{2}\frac{\|W_{\mathrm{kq}}^{(t)}\|}{\|W_{\mathrm{kq}}^*\|} \right) \|W_{\mathrm{kq}}^*\| \\
&\geq 2L_{\max} \exp\left( -\frac{\beta_0}{2}\frac{\|W_{\mathrm{kq}}^{(t)}\|}{\|W_{\mathrm{kq}}^*\|} \right) \|W_{\mathrm{kq}}^*\| A^{(n,*)} \\
&\overset{(c)}{\geq} \frac{\|W_{\mathrm{kq}}^*\|}{\|W_{\mathrm{kq}}^{(t)}\|} A^{(n,*)} \\
&\geq \beta_0 A^{(n,*)},
\end{aligned}
$$

where $(a)$ follows from that $b_{*,\ell} \leq -(1+\beta_0)\|W_{\mathrm{kq}}^{(t)}\|/\|W_{\mathrm{kq}}^*\|$ and Equation (11), and $(b)$ is due to the fact that $\sum_{i \in [|\mathcal{V}|]}[\mathrm{T}_\theta^{(n)}]_i = 1$ for any $\theta, n$, and $(c)$ follows from $\beta_0/2 \geq \frac{\|W_{\mathrm{kq}}^*\|}{\|W_{\mathrm{kq}}^{(t)}\|}\log(2L_{\max}\|W_{\mathrm{kq}}^{(t)}\|)$

For the term $B_-^{(n,t)}$, we have

$$
\begin{aligned}
B_-^{(n,t)} &= \sum_{\ell \notin l(n)} [\mathrm{T}_{\theta^{(t)}}^{(n)}]_{\mathrm{In}(x_\ell^{(n)})} \varphi_\ell^{(n,t)} \left( \varphi_-^{(n,t)} a_\ell^{(n,t)} - \sum_{\ell' \notin l(n)} \varphi_{\ell'}^{(n,t)} a_{\ell'}^{(n,t)} \right) \\
&= \sum_{\ell \notin l(n)} [\mathrm{T}_{\theta^{(t)}}^{(n)}]_{\mathrm{In}(x_\ell^{(n)})} \varphi_\ell^{(n,t)} \varphi_-^{(n,t)} \left( a_\ell^{(n,t)} - \sum_{\ell' \notin l(n)} \frac{\varphi_{\ell'}^{(n,t)}}{\varphi_-^{(n,t)}} a_{\ell'}^{(n,t)} \right) \\
&= \max_{\mathrm{T}>0} \left\{ \sum_{\substack{\ell \notin l(n) \\ \mathrm{diff}>-\mathrm{T}}} [\mathrm{T}_{\theta^{(t)}}^{(n)}]_{\mathrm{In}(x_\ell^{(n)})} \varphi_\ell^{(n,t)} \varphi_-^{(n,t)} \left( \underbrace{a_\ell^{(n,t)} - \sum_{\ell' \notin l(n)} \frac{\varphi_{\ell'}^{(n,t)}}{\varphi_-^{(n,t)}} a_{\ell'}^{(n,t)}}_{\mathrm{diff}} \right) \right. \\
&\qquad\qquad \left. + \sum_{\substack{\ell \notin l(n) \\ \mathrm{diff}<-\mathrm{T}}} [\mathrm{T}_{\theta^{(t)}}^{(n)}]_{\mathrm{In}(x_\ell^{(n)})} \varphi_\ell^{(n,t)} \varphi_-^{(n,t)} \left( \underbrace{a_\ell^{(n,t)} - \sum_{\ell' \notin l(n)} \frac{\varphi_{\ell'}^{(n,t)}}{\varphi_-^{(n,t)}} a_{\ell'}^{(n,t)}}_{\mathrm{diff}} \right) \right\}
\end{aligned}
$$

$$\overset{(a)}{\geq} \max_{\mathtt{T}>0} \left\{ \sum_{\ell \notin l(n)} [\mathtt{T}^{(n)}_{\theta^{(t)}}]_{\mathrm{In}(x^{(n)}_\ell)} \varphi^{(n,t)}_- \left( -\varphi^{(n,t)}_\ell \mathtt{T} - 2\|W^{(t)}_{\mathrm{kq}}\| \exp(-\mathtt{T}) \right) \right\}$$

$$\overset{(b)}{\geq} -L_{\max}(1 - [\mathtt{T}^{(n)}_{\theta^{(t)}}]_{\mathrm{In}(x^{(n)}_{\ell_*})}) \varphi^{(n,t)}_- \left( 1 + \log(2\|W^{(t)}_{\mathrm{kq}}\|) \right)$$

$$\geq \frac{L_{\max}\|W^*_{\mathrm{kq}}\|}{\varphi^{(n,t)}_+ \|W^{(t)}_{\mathrm{kq}}\|} \left( 1 + \log(2\|W^{(t)}_{\mathrm{kq}}\|) \right) A^{(n,*)}$$

$$\overset{(c)}{\geq} \frac{L^2_{\max}\|W^*_{\mathrm{kq}}\|}{\|W^{(t)}_{\mathrm{kq}}\|} \left( 1 + \log(2\|W^{(t)}_{\mathrm{kq}}\|) \right) A^{(n,*)}$$

$$\overset{(d)}{\geq} \beta_0 A^{(n,*)},$$

where $(a)$ follows from Equation (11), $(b)$ is optained by choosing $\mathtt{T} = \log(2\|W^{(t)}_{\mathrm{kq}}\|)$, $(c)$ follows from Lemma 5, and $(d)$ is due to the fact that $\beta_0 \geq \frac{L^2_{\max}\|W^*_{\mathrm{kq}}\|}{\|W^{(t)}_{\mathrm{kq}}\|}(1 + \log(2\|W^{(t)}_{\mathrm{kq}}\|))$.

So far, we have shown that for if $\varphi^{(n,t)}_- \geq \exp(-(1 + \beta_0/2)\|W^{(t)}_{\mathrm{kq}}\|/\|W^*_{\mathrm{kq}}\|)$, then

$$A^{(n,t)} + B^{(n,t)} = A^{(n,t)} + B^{(n,t)}_+ + B^{(n,t)}_-$$
$$\geq (1 + \beta_0)A^{(n,*)} + \left( (1 + \beta_0)B^{(n,*)} + \beta_0 A^{(n,*)} \right) + \beta_0 A^{(n,*)}$$
$$= (1 + 3\beta_0)A^{(n,*)} + (1 + \beta_0)B^{(n,*)}$$
$$\geq (1 + 3\beta_0)(A^{(n,*)} + B^{(n,*)}).$$

**Type 2.** Now consider sentence $X^{(n)}$ such that $\varphi^{(n,t)}_- < \exp(-(1 + \beta_0/2)\|W^{(t)}_{\mathrm{kq}}\|/\|W^*_{\mathrm{kq}}\|)$.

Let $n_0$ be the type 1 sample such that $\varphi^{(n_0,t)}_- \geq \exp(-\|W^{(t)}_{\mathrm{kq}}\|/\|W^*_{\mathrm{kq}}\|)$

Then, we aim to show that

$$A^{(n,t)} + B^{(n,t)} \geq \beta_0 A^{(n_0,*)}$$

Note that

$$A^{(n,t)} \geq \sum_{\ell_* \in l(n)} ([\mathtt{T}^{(n)}_{\theta^{(t)}}]_{\mathrm{In}(x^{(n)}_{\ell_*})} - 1)\varphi^{(n,t)}_{\ell_*}\varphi^{(n,t)}_- \left( 1 + \log\frac{L_{\max}\|W^{(t)}_{\mathrm{kq}}\|}{\varphi^{(n,t)}_-} \right)$$

$$\geq ([\mathtt{T}^{(n)}_{\theta^{(t)}}]_{\mathrm{In}(x^{(n)}_{\ell_*})} - 1)\exp\left( -(1 + \beta_0/2)\frac{\|W^{(t)}_{\mathrm{kq}}\|}{\|W^*_{\mathrm{kq}}\|} \right) \left( 1 + (1 + \beta_0/2)\frac{\|W^{(t)}_{\mathrm{kq}}\|}{\|W^*_{\mathrm{kq}}\|} + \log(L_{\max}\|W^{(t)}_{\mathrm{kq}}\|) \right)$$

$$\geq (1 + \beta_0)([\mathtt{T}^{(n)}_{\theta^{(t)}}]_{\mathrm{In}(x^{(n)}_{\ell_*})} - 1)\exp\left( -(1 + \beta_0/2)\frac{\|W^{(t)}_{\mathrm{kq}}\|}{\|W^*_{\mathrm{kq}}\|} \right) \frac{\|W^{(t)}_{\mathrm{kq}}\|}{\|W^*_{\mathrm{kq}}\|}$$

and

$$B^{(n,t)} = \sum_{\ell \notin l(n)} [\mathtt{T}^{(n)}_{\theta^{(t)}}]_{\mathrm{In}(x^{(n)}_\ell)} \varphi^{(n,t)}_\ell \left( a^{(n,t)}_\ell - \sum_{\ell'} \varphi^{(n,t)}_{\ell'} a^{(n,t)}_{\ell'} \right)$$

$$\geq -2(1 - [\mathtt{T}^{(n)}_{\theta^{(t)}}]_{\mathrm{In}(x^{(n)}_{\ell_*})})\exp\left( -(1 + \beta_0/2)\frac{\|W^{(t)}_{\mathrm{kq}}\|}{\|W^*_{\mathrm{kq}}\|} \right) \|W^{(t)}_{\mathrm{kq}}\|$$

Since

$$A^{(n_0,*)} \leq ([T^{(n_0)}_{\theta^{(t)}}]_{\text{In}(x^{(n_0)}_{\ell_*})} - 1)\varphi_+^{(n_0,t)}\varphi_-^{(n_0,t)}\frac{\|W^{(t)}_{\text{kq}}\|}{\|W^*_{\text{kq}}\|}$$

$$\leq ([T^{(n_0)}_{\theta^{(t)}}]_{\text{In}(x^{(n_0)}_{\ell_*})} - 1)\varphi_+^{(n_0,t)}\exp\left(-\frac{\|W^{(t)}_{\text{kq}}\|}{\|W^*_{\text{kq}}\|}\right)\frac{\|W^{(t)}_{\text{kq}}\|}{\|W^*_{\text{kq}}\|}$$

We further note that $[T^{(n_0)}_{\theta^{(t)}}]_{\text{In}(x^{(n_0)}_{\ell_*})} < [T^{(n)}_{\theta^{(t)}}]_{\text{In}(x^{(n)}_{\ell_*})}$ due to $\varphi_+^{(n_0,t)} < \varphi_+^{(n,t)}$. Thus,

$$A^{(n,t)} + B^{(n,t)}$$

$$\geq \exp\left(-\beta_0/2\frac{\|W^{(t)}_{\text{kq}}\|}{\|W^*_{\text{kq}}\|}\right)(1 + \beta_0 + 2\|W_{\text{kq}^*}\|)\frac{A^{(n_0,*)}}{\varphi_+^{(n_0,t)}}$$

$$\overset{(a)}{\geq} \beta_0 A^{(n_0,*)},$$

where $(a)$ is due to that $\beta_0 \geq \frac{2\|W^*_{\text{kq}}\|(1+2\|W^*_{\text{kq}}\|)}{\|W^{(t)}_{\text{kq}}\|}\log(1 + \frac{\|W^{(t)}_{\text{kq}}\|}{2\|W^*_{\text{kq}}\|})$, $\|W^{(t)}_{\text{kq}}\| \geq 2(e-1)\|W^*_{\text{kq}}\|$, and

$$\beta_0 \geq \frac{2\|W^*_{\text{kq}}\|}{\|W^{(t)}_{\text{kq}}\|}\log\frac{1 + \beta_0 + 2\|W^*_{\text{kq}}\|}{\beta_0}.$$

In summary, we have that

$$\sum_n \pi^{(n)}(A^{(n,t)} + B^{(n,t)})$$

$$= \sum_{n\text{ is type 2}} \pi^{(n)}(A^{(n,t)} + B^{(n,t)}) + \sum_{n\text{ is type 1}} \pi^{(n)}(A^{(n,t)} + B^{(n,t)})$$

$$\geq \max_{n_0\text{ is type 1}} \beta_0 A^{(n_0,*)} + \sum_{n\text{ is type 1}} \pi^{(n)}((1 + 3\beta_0)A^{(n,*)} + (1 + \beta_1)B^{(n,*)})$$

$$\geq \sum_{n\text{ is type 1}} N\pi^{(n)}\beta_0(A^{(n,*)} + B^{(n,*)}) + (1 + 3\beta_0)\sum_{n\text{ is type 1}} \pi^{(n)}(A^{(n,*)} + B^{(n,*)})$$

$$\geq (1 + (N+3)\beta_0)\sum_{n\text{ is type 1}} \pi^{(n)}(A^{(n,*)} + B^{(n,*)})$$

$$\geq (1 + \alpha_t)\sum_{n\text{ is type 2}} \pi^{(n)}(A^{(n,*)} + B^{(n,*)}) + (1 + \alpha_t)\sum_{n\text{ is type 1}} \pi^{(n)}(A^{(n_0,*)} + B^{(n_0,*)})$$

$$= (1 + \alpha_t)\sum_n \pi^{(n)}(A^{(n,*)} + B^{(n,*)}),$$

where $\alpha_t \geq (N+3)\beta_0$. The proof is finished.

∎

Now, we are ready to prove Theorem 1.

### C.5 Proof of Theorem 1

*Proof of Theorem 1.*

Recall that $\alpha_t = \frac{4NL^2_{\max}\|W^*_{\text{kq}}\|^2}{\|W^{(t)}_{\text{kq}}\|}\left(1 + \log\left(2L_{\max}\|W^{(t)}_{\text{kq}}\|\right)\right)$. By Lemma 7, we have

$$\left\langle W^{(t+1)}_{\text{kq}} - W^{(t)}_{\text{kq}}, \frac{W^*_{\text{kq}}}{\|W^*_{\text{kq}}\|}\right\rangle$$

$$
= -\eta \left\langle \nabla_{W_{\mathrm{kq}}} \mathcal{L}(\theta^{(t)}), \frac{W_{\mathrm{kq}}^*}{\|W_{\mathrm{kq}}^*\|} \right\rangle
$$

$$
\geq -\frac{\eta}{1+\alpha_t} \left\langle \nabla_{W_{\mathrm{kq}}} \mathcal{L}(\theta^{(t)}), \frac{W_{\mathrm{kq}}^{(t)}}{\|W_{\mathrm{kq}}^{(t)}\|} \right\rangle
$$

$$
= \frac{1}{1+\alpha_t} \left\langle W_{\mathrm{kq}}^{(t+1)} - W_{\mathrm{kq}}^{(t)}, \frac{W_{\mathrm{kq}}^{(t)}}{\|W_{\mathrm{kq}}^{(t)}\|} \right\rangle
$$

$$
= \frac{1}{2\|W_{\mathrm{kq}}^{(t)}\|} \left( \|W_{\mathrm{kq}}^{(t+1)}\|^2 - \|W_{\mathrm{kq}}^{(t+1)} - W_{\mathrm{kq}}^{(t)}\|^2 - \|W_{\mathrm{kq}}^{(t)}\|^2 \right) - \frac{\alpha_t}{1+\alpha_t} \left\langle W_{\mathrm{kq}}^{(t+1)} - W_{\mathrm{kq}}^{(t)}, \frac{W_{\mathrm{kq}}^{(t)}}{\|W_{\mathrm{kq}}^{(t)}\|} \right\rangle
$$

$$
= \frac{\|W_{\mathrm{kq}}^{(t)}\|^2 - \|W_{\mathrm{kq}}^{(t)}\|^2}{2\|W_{\mathrm{kq}}^{(t)}\|} - \frac{\eta^2}{2\|W_{\mathrm{kq}}^{(t)}\|} + \frac{\eta \alpha_t}{1+\alpha_t} \left\langle \frac{\nabla_{W_{\mathrm{kq}}} \mathcal{L}(\theta^{(t)})}{\|\nabla_{W_{\mathrm{kq}}} \mathcal{L}(\theta^{(t)})\|}, \frac{W_{\mathrm{kq}}^{(t)}}{\|W_{\mathrm{kq}}^{(t)}\|} \right\rangle
$$

$$
\geq \|W_{\mathrm{kq}}^{(t+1)}\| - \|W_{\mathrm{kq}}^{(t)}\| - \frac{\eta^2}{2\|W_{\mathrm{kq}}^{(t)}\|} - \frac{\eta \alpha_t}{1+\alpha_t}.
$$

Let $t_0 = \lceil \frac{8 L_{\max} \|W_{\mathrm{kq}}^*\|^2}{\eta} \rceil$ be defined in Lemma 7. Summing over $t$ from $t_0$, we have

$$
\left\langle W_{\mathrm{kq}}^{(t)} - W_{\mathrm{kq}}^{(t_0)}, \frac{W_{\mathrm{kq}}^*}{\|W_{\mathrm{kq}}^*\|} \right\rangle \geq \|W_{\mathrm{kq}}^{(t)}\| - \|W_{\mathrm{kq}}^{(t_0)}\| - \sum_{t'=t_0}^{t-1} \frac{\eta^2}{2\|W_{\mathrm{kq}}^{(t')}\|} - \sum_{t'=t_0}^{t-1} \frac{\eta \alpha_{t'}}{1+\alpha_{t'}}
$$

By Lemma 6, we have

$$
\sum_{t'=t_0}^{t-1} \frac{1}{\|W_{\mathrm{kq}}^{(t')}\|} \leq \sum_{t'=t_0}^{t-1} \frac{2 L_{\max} \|W_{\mathrm{kq}}^*\|/\eta}{t'}
$$

$$
\leq \frac{2 L_{\max} \|W_{\mathrm{kq}}^*\|}{\eta} \log t.
$$

Furthermore,

$$
\sum_{t'=t_0}^{t-1} \frac{\alpha_{t'}}{1+\alpha_{t'}} \leq \sum_{t'=t_0}^{t-1} \alpha_{t'}
$$

$$
= \sum_{t'=t_0}^{t-1} \frac{4 N L_{\max}^2 \|W_{\mathrm{kq}}^*\|^2}{\|W_{\mathrm{kq}}^{(t')}\|} \left( 1 + \log \left( 2 L_{\max} \|W_{\mathrm{kq}}^{(t')}\| \right) \right)
$$

$$
= \sum_{t'=t_0}^{t-1} \frac{4 N L_{\max}^2 \|W_{\mathrm{kq}}^*\|^2}{\|W_{\mathrm{kq}}^{(t')}\|} \left( 1 + \log \left( 2 L_{\max} \right) \right) + \sum_{t'=t_0}^{t-1} \frac{4 N L_{\max}^2 \|W_{\mathrm{kq}}^*\|^2}{\|W_{\mathrm{kq}}^{(t')}\|} \log \|W_{\mathrm{kq}}^{(t')}\|
$$

$$
\leq \sum_{t'=t_0}^{t-1} \frac{8 N L_{\max}^3 \|W_{\mathrm{kq}}^*\|^3/\eta}{t'} \log(2 e L_{\max}) + \sum_{t'=t_0}^{t-1} \frac{8 N L_{\max}^3 \|W_{\mathrm{kq}}^*\|^3/\eta}{t'} \log \frac{t'}{2 L_{\max} \|W_{\mathrm{kq}}^*\|/\eta}
$$

$$
= \sum_{t'=t_0}^{t-1} \frac{8 N L_{\max}^3 \|W_{\mathrm{kq}}^*\|^3/\eta}{t'} \log(e\eta/\|W_{\mathrm{kq}}^*\|) + \sum_{t'=t_0}^{t-1} \frac{8 N L_{\max}^3 \|W_{\mathrm{kq}}^*\|^3/\eta}{t'} \log(t')
$$

$$
\overset{(a)}{\leq} \frac{8 N L_{\max}^3 \|W_{\mathrm{kq}}^*\|^3}{\eta} \log^2 t,
$$

where $(a)$ follows from $\eta \leq \|W_{\mathrm{kq}}^*\|/e$.

Therefore, we have

$$
\left\langle W_{\mathrm{kq}}^{(t)} - W_{\mathrm{kq}}^{(t_0)}, \frac{W_{\mathrm{kq}}^*}{\|W_{\mathrm{kq}}^*\|} \right\rangle \geq \|W_{\mathrm{kq}}^{(t)}\| - \|W_{\mathrm{kq}}^{(t_0)}\| - \sum_{t'=t_0}^{t-1} \frac{\eta^2}{2\|W_{\mathrm{kq}}^{(t')}\|} - \sum_{t'=t_0}^{t-1} \frac{\eta \alpha_{t'}}{1+\alpha_{t'}}
$$

$$\geq \|W_{\mathrm{kq}}^{(t)}\| - \|W_{\mathrm{kq}}^{(t_0)}\| - L_{\max}\|W_{\mathrm{kq}}^*\| \log t - 8NL_{\max}^3\|W_{\mathrm{kq}}^*\|^3 \log^2 t.$$

Finally, by Lemma 6, and $t_0 \leq 1 + 8L_{\max}\|W_{\mathrm{kq}}^*\|^2/\eta$, we have $\|W_{\mathrm{kq}}^{(t_0)}\| \leq 9L_{\max}\|W_{\mathrm{kq}}^*\|^2$, and

$$\left\langle \frac{W_{\mathrm{kq}}^{(t)}}{\|W_{\mathrm{kq}}^{(t)}\|}, \frac{W_{\mathrm{kq}}^*}{\|W_{\mathrm{kq}}^*\|} \right\rangle \geq 1 - \frac{2\|W_{\mathrm{kq}}^{(t_0)}\| + L_{\max}\|W_{\mathrm{kq}}^*\| \log t + 8NL_{\max}^3\|W_{\mathrm{kq}}^*\|^3 \log^2 t}{\|W_{\mathrm{kq}}^{(t)}\|}$$

$$\geq 1 - \frac{54NL_{\max}^4\|W_{\mathrm{kq}}^*\|^4 \log^2 t}{t\eta}.$$

The proof is finished.

∎

### C.6  Proof of Theorem 2

*Proof of Theorem 2.*

Recall that $T \geq 384\|W_{\mathrm{ov}}^*\|^5 \log(2|\mathcal{V}|) \log T/\eta_0$, and $\Delta = T\eta_0/(4\|W_{\mathrm{ov}}^*\|^2)$ due to Corollary 1 and

$$(e_{\mathrm{In}(x)} - e_v)^\top W_{\mathrm{ov}}^{(T)} x = \frac{T\eta_0}{4\|W_{\mathrm{ov}}^*\|^2}$$

Note that for all $\ell_* \in l(n)$ and $\ell \notin l(n)$

$$(x_\ell^{(n)} - x_{\ell*}^{(n)})^\top W_{\mathrm{kq}}^{(t)} X_{-1}^{(n)}$$

$$= \frac{\|W_{\mathrm{kq}}^{(t)}\|}{\|W_{\mathrm{kq}}^*\|}(x_\ell^{(n)} - x_{\ell*}^{(n)})^\top W_{\mathrm{kq}}^* X_{-1}^{(n)} + (x_\ell^{(n)} - x_{\ell*}^{(n)})^\top \left( W_{\mathrm{kq}}^{(t)} - \frac{\|W_{\mathrm{kq}}^{(t)}\|}{\|W_{\mathrm{kq}}^*\|}W_{\mathrm{kq}^*} \right) X_{-1}^{(n)}$$

$$\leq -\frac{\|W_{\mathrm{kq}}^{(t)}\|}{\|W_{\mathrm{kq}}^*\|} + 2\|W_{\mathrm{kq}}^{(t)}\| \left\| \frac{W_{\mathrm{kq}}^{(t)}}{\|W_{\mathrm{kq}}^{(t)}\|} - \frac{W_{\mathrm{kq}}^*}{\|W_{\mathrm{kq}}^*\|} \right\| \tag{13}$$

$$\leq -\frac{t\eta}{2L_{\max}\|W_{\mathrm{kq}}^*\|^2} + \frac{\sqrt{2}t\eta}{L_{\max}\|W_{\mathrm{kq}}^*\|}\sqrt{\frac{54NL_{\max}^4\|W_{\mathrm{kq}}^*\|^4 \log^2 t}{t\eta}}$$

$$\overset{(a)}{\leq} -\frac{t\eta}{4L_{\max}\|W_{\mathrm{kq}}^*\|^2},$$

where $(a)$ follows from $t \geq 1696NL_{\max}^4\|W_{\mathrm{kq}}^*\|^6 \log^2 t/\eta$.

Therefore,

$$\sum_{\ell_* \in l(n)} \varphi_{\ell*}^{(n,t)} = \frac{|l(n)|}{|l(n)| + \sum_{\ell' \notin l(n)} \exp\left( (x_\ell^{(n)} - x_{\ell*}^{(n)})^\top W_{\mathrm{kq}}^{(t)} X_{-1}^{(n)} \right)}$$

$$\geq \frac{|l(n)|}{|l(n)| + (L^{(n)} - |l(n)|) \exp\left( -\frac{t\eta}{4L_{\max}\|W_{\mathrm{kq}}^*\|^2} \right)}$$

$$\geq \frac{1}{1 + L_{\max} \exp\left( -\frac{t\eta}{4L_{\max}\|W_{\mathrm{kq}}^*\|^2} \right)}$$

$$\geq \frac{1}{1 + \epsilon},$$

where the last inequality follows from $t \geq \frac{4L_{\max}\|W_{\mathrm{kq}}^*\|}{\eta} \log \frac{L_{\max}}{\epsilon}$.

Hence, the loss on the sentence $X^{(n)}$ satisfies that

$$-\log\left( e_{\mathrm{In}(X^{(n)})}^\top \mathrm{T}_{\theta^{(t)}}(X^{(n)}) \right)$$

$$= -\log \frac{\exp\left(e_{\mathrm{In}(X^{(n)})}^\top W_{\mathrm{ov}}^{(T)} \sum_\ell x_\ell^{(n)} \varphi_\ell^{(n,t)}\right)}{\sum_{v \le |\mathcal{V}|} \exp\left(e_v^\top W_{\mathrm{ov}}^{(T)} \sum_\ell x_\ell^{(n)} \varphi_\ell^{(n,t)}\right)}$$

$$= -\log \frac{1}{1 + \sum_{v \ne \mathrm{In}(X^{(n)})} \exp\left((e_v - e_{\mathrm{In}(X^{(n)})})^\top W_{\mathrm{ov}}^{(T)} \sum_\ell x_\ell^{(n)} \varphi_\ell^{(n,t)}\right)}$$

$$= \log\left(1 + \sum_{v \ne \mathrm{In}(X^{(n)})} \exp\left((e_v - e_{\mathrm{In}(X^{(n)})})^\top W_{\mathrm{ov}}^{(T)} \sum_{\ell \notin l(n)} x_\ell^{(n)} \varphi_\ell^{(n,t)}\right)\right)$$

$$= \log\left(1 + \sum_{v \ne \mathrm{In}(X^{(n)})} \exp\left(-\Delta \sum_{\ell_* \in l(n)} \varphi_{\ell_*}^{(n,t)} + \Delta \sum_{\ell \notin l(n)} \varphi_\ell^{(n,t)}\right)\right)$$

$$\le |\mathcal{V}| \exp\left(-\Delta(2\varphi_+^{(n,t)} - 1)\right)$$

$$\le |\mathcal{V}| \exp\left(-\Delta + \frac{2\Delta L_{\max}}{L_{\max} + \exp\left(\frac{t\eta}{4L_{\max}\|W_{\mathrm{kq}}^*\|^2}\right)}\right).$$

Thus, the average loss has upper bound, which is

$$|\mathcal{V}| \exp\left(-\Delta + \frac{2\Delta L_{\max}}{L_{\max} + \exp(C_1 t)}\right),$$

for $C_1 = \eta/(4L_{\max}\|W_{\mathrm{kq}}^*\|^2)$, and $\Delta = C_0 T$ for $C_0 = \eta_0/(4\|W_{\mathrm{ov}}^*\|^2)$.

∎

# D  Proof of Proposition 2 and Theorem 3

## D.1  Proof of Proposition 2

**Proposition 4 (Restatement of Proposition 2)**  *Under Assumption 2, if $W_{\mathrm{kq}}^*$ satisfies Equation (3), i.e.,*

$$W_{\mathrm{kq}}^* = \arg\min \|W\|, \quad s.t. \quad (x_{\ell_*}^{(n)} - x_\ell^{(n)})^\top W x \ge 1, \quad \forall \ell \notin l(n), \forall n.$$

*In addition, for each query $x^q$, if there are $k$ optimal tokens under a $x^q$-partial order, $m$ non-optimal tokens under $x^q$-partial order, then, for any optimal token $x_*$, non-optimal token $x$, and non-comparable token $x_0$, we have*

$$x_*^\top W_{\mathrm{kq}}^* x^q = \frac{m}{k+m}, \quad x^\top W_{\mathrm{kq}}^* x^q = -\frac{k}{k+m}, \quad x_0^\top W_{\mathrm{kq}}^* x^q = 0.$$

*A direct result is that*

$$(x_\ell^{(n)} - x_{\ell'}^{(n)})^\top W_{\mathrm{kq}}^* X_{-1}^{(n)} = 0, \quad \forall \ell, \ell' \notin l(n).$$

*Proof.*  Let $U \in \mathbb{R}^d$ be the rotation matrix such that $Ux = e_{\mathrm{I}(x)}$. Because $U$ preserves Frobenius norm, the optimization problem in Equation (3) can be written as

$$\tilde{W}^* = \arg\min \|W\|, \quad \text{s.t.} (e_{\mathrm{I}(x_{\ell_*}^{(n)})} - e_{\mathrm{I}(x_\ell^{(n)})})^\top W e_{\mathrm{I}(X_{-1}^{(n)})} \ge 1. \tag{14}$$

Notably $\tilde{W}^* = U W_{\mathrm{kq}}^* U^\top$.

Note that $\{e_{\mathrm{I}(X_{-1}^{(n)})}\}_n$ forms a standard basis. It suffices to minimize the norm of each column of $W$ subject to the constraint $(e_{\mathrm{I}(x_{\ell_*}^{(n)})} - e_{\mathrm{I}(x_\ell^{(n)})})^\top W e_{\mathrm{I}(X_{-1}^{(n)})} \ge 1$.

Let us consider any column $c$ of $W$, denoted as $[w_1, \ldots, w_d]^\top$. Without loss of generality, we assume that, for all $X^{(n)}$ with $\mathrm{I}(X_{-1}^{(n)}) = c$, the set of indices of the optimal tokens of those samples are

$\{1, \ldots, k\}$, and the set of indices of the non-optimal tokens are $\{k + 1, \ldots, k + m\}$. Then, the optimization problem Equation (14) reduces to the following problem

$$\min w_1^2 + \ldots + w_d^2, \quad \text{s.t.} \quad w_i - w_j \geq 1, \quad \forall i \leq k, j \in A_i \subset \{k+1, \ldots, k+m\}, \quad (15)$$

where $A_i$ is the set of indices of the non-optimal tokens in some samples whose optimal token has index $i$.

In other words, each column of the solution of Equation (14) is the solution of Equation (15).

Note that Equation (15) is a convex problem with linear constraints. The Lagrangian function is

$$L(\lambda) = \sum_{i=1}^{k+m} w_i^2 + 2 \sum_{i=1}^{m} \sum_{j \in A_i} \lambda_{ij}(1 - w_i + w_j),$$

where we directly set $w_j = 0$ for all $j \in \{k + m + 1, \ldots, d\}$. That is, non-comparable tokens have value 0.

By KKT-condition, we have

$$\begin{cases} w_i = \sum_{j \in A_i} \lambda_{ij}, & \forall i \leq k \\ w_j = -\sum_{i=1}^{k} \lambda_{ij} \mathbb{1}\{j \in A_i\}, & \forall k+1 \leq j \leq k+m \end{cases}$$

Thus,

$$\min w_1^2 + \ldots + w_d^2$$

$$= \max_\lambda \left\{ -\sum_{i=1}^{k} \left( \sum_{j \in A_i} \lambda_{ij} \right)^2 - \sum_{j=k+1}^{k+m} \left( \sum_{i=1}^{k} \lambda_{ij} \mathbb{1}\{j \in A_i\} \right)^2 + 2 \sum_{i=1}^{m} \sum_{j \in A_i} \lambda_{ij} \right\}$$

Let

$$L^*(\lambda) = -\sum_{i=1}^{k} \left( \sum_{j \in A_i} \lambda_{ij} \right)^2 - \sum_{j=k+1}^{k+m} \left( \sum_{i=1}^{k} \lambda_{ij} \mathbb{1}\{j \in A_i\} \right)^2 + 2 \sum_{i=1}^{m} \sum_{j \in A_i} \lambda_{ij},$$

where $\lambda \geq 0$. The maximum of $L^*$ is achieved when $\nabla_\lambda L^* = 0$. This implies that

$$\sum_{j \in A_{i_0}} \lambda_{i_0 j} + \sum_{i=1}^{k} \lambda_{ij} \mathbb{1}\{j_0 \in A_i\} = 1, \quad \forall 1 \leq i_0 \leq k < j_0 \leq k+m.$$

Hence, we have $w_i - w_j = 1$ for all $1 \leq i \leq k < j \leq k + m$, which means the optimum of the original problem is achieved on the boundary. Therefore, we reduce the original problem to

$$\min_x k(x+1)^2 + mx^2,$$

where $x = w_{k+1} = \ldots = w_{k+m}$. Hence, the optimal solution is $w_1 = \ldots = w_k = m/(m+k)$, and $w_{k+1} = \ldots = w_{k+m} = -k/(k+m)$.

Therefore, the solution of Equation (15) satisfies that the "optimal values" are the same and the "non-optimal values" are the same as well. This fact proves that

$$(x_\ell^{(n)} - x_{\ell'}^{(n)}) W_{kq}^* X_{-1}^{(n)} = 0, \forall \ell, \ell' \notin l(n).$$

And moreover, if there are $k$ optimal tokens under a $x^q$-partial order, $m$ non-optimal tokens under $x^q$-partial order, then, for any optimal token $x_*$ and non-optimal token $x$, we have

$$x_*^\top W_{kq}^* x^q = \frac{m}{k+m}, \quad x^\top W_{kq}^* x^q = -\frac{k}{k+m}.$$

∎

## D.2 Proof of Theorem 3

*Proof.* The proof follows similar logic to Theorem 2. By Equation (13), we have for any $x, x' \in \mathcal{V}$

$$(x - x')^\top W_{\text{kq}}^{(t)} x^q$$

$$\geq \frac{\|W_{\text{kq}}^{(t)}\|}{\|W_{\text{kq}}^*\|}(x - x')^\top W_{\text{kq}}^* x^q - \frac{\sqrt{2}t\eta}{L_{\max}\|W_{\text{kq}}^*\|}\sqrt{\frac{54NL_{\max}^4\|W_{\text{kq}}^*\|^4\log^2 t}{t\eta}}$$

$$\overset{(a)}{\geq} \frac{t\eta}{2L_{\max}\|W_{\text{kq}}^*\|^2}(x - x')^\top W_{\text{kq}}^* x^q - \sqrt{108t\eta NL_{\max}^2\|W_{\text{kq}}^*\|^2\log^2 t},$$

where $(a)$ follows from Lemma 6. The first part of Theorem 3 follows from Theorem 3.

For the second part of Theorem 3, Let $X = [x_1, \ldots, x_L]$ such that for $\ell_0 \in l_0 \subset \{1, \ldots, L\}$, $x_{\ell_0} = x$ is a non-comparable token, and other tokens are non-optimal under the $x_L$-partial order.

Let $\varphi_\ell \propto \exp(x_\ell W_{\text{kq}}^{(t)} x_L)$ for sufficiently large $t = \Omega(\log(1/\epsilon))$ such that $\sum_{\ell_0 \in l_0} \varphi_{\ell_0} \geq 1 - \epsilon$.

Then, we have

$$e_{\text{In}(x)}^\top T_{\theta^{(t)}}(X) = \frac{\exp\left(e_{\text{In}(x)}^\top W_{\text{ov}}^{(T)} \sum_\ell x_\ell \varphi_\ell\right)}{\sum_{v \leq |\mathcal{V}|} \exp\left(e_v^\top W_{\text{ov}}^{(T)} \sum_\ell x_\ell \varphi_\ell\right)}$$

$$= \frac{1}{1 + \sum_{v \neq \text{In}(x)} \exp\left((e_v - e_{\text{In}(x)})^\top W_{\text{ov}}^{(T)} \sum_\ell x_\ell \varphi_\ell\right)}$$

$$= \frac{1}{1 + \sum_{v \neq \text{In}(x)} \exp\left(-\Delta \sum_{\ell_0 \in l_0} \varphi_{\ell_0} + \Delta \sum_{\ell \notin l_0} \varphi_\ell\right)}$$

$$\geq \frac{1}{1 + |\mathcal{V}|\exp\left(-\Delta(1 - 2\epsilon)\right)}$$

$$\geq 1 - \epsilon_0,$$

where the last inequality follows from $T = O(\log(1/\epsilon_0))$. Therefore, the trained transformer will predict $n(x)$, the next token of the non-comparable token.

∎

