# OpenReview forum: "Non-asymptotic Convergence of Training Transformers for Next-token Prediction"
_NeurIPS.cc/2024/Conference — NeurIPS 2024 poster_

### Official Review · Reviewer_jsSk · 2024-07-08

**Soundness:** 3
**Presentation:** 3
**Contribution:** 3
**Rating:** 6
**Confidence:** 3

**Summary:**

This manuscript focuses on the next-token prediction (NTP) task, and provides a fine-grained non-asymptotic analysis of the training dynamics of a one-layer transformer. Specifically, the authors first characterize the essential structural properties of training datasets for NTP using a mathematical framework based on partial orders. Then, they design a two-stage training algorithm, where the pre-processing stage for training the feed-forward layer and the main stage for training the attention layer exhibit fast convergence performance. Finally, they show that the well-trained transformer can have non-trivial prediction ability on unseen data, which sheds light on the generalization capability of transformers.

**Strengths:**

This manuscript conducts a theoretical analysis on the convergence speed and generalization of the important NTP task, with giving an example to illustrate. Although not comprehensive, it can serve as a stepping stone for convergence analysis of NTP tasks.
This manuscript is well-written and easy to read.

**Weaknesses:**

1. How is the loss function between lines 227 and 288 obtained? Why is its form different from Eq. 1?
2. In Section 6, the authors theoretically prove the generalization ability of the trained Transformer. Can the generalization ability be proved in the experimental part?
3. Why do the authors design a two-stage training algorithm? Is it valuable in practical applications compared to single-stage training?

**Questions:**

Please see the weakness

**Limitations:**

This manuscript simplifies the Transformer into a single layer for analysis. While I understand the need for this simplification, it would be nice to include a brief paragraph outlining the views on generalizing to multi-layer Transformers.

---

> ### Author Rebuttal · Authors · 2024-08-07
>
> We thank the reviewer for the helpful comments, and would like to provide the following responses.
>
> **Q1:** How is the loss function between lines 227 and 228 obtained? Why is its form different from Eq. 1?
>
> **A1:** The loss in Eq.\ (1) is for the general-length case. The loss between lines 227 and 228 is for the training stage 1 (i.e., training of feed-forward layer), where the inputs are one-token sentences. We note that there was a typo in Eq.\ (1), i.e., missing $\exp$ in the first bracket. The correct form should be
> \\[ \log \left(\sum_v \exp( e_v^\top \bar{T}(X) ) \right) - e_i^\top \bar{T}(X) ,\\]
> where $i$ is the index of the next token of $X$. Such a form can specialize to the loss for stage 1 as follows. Let $X = [x_1]$. Then we have $\bar{T}(X) = W_{ov}x_1$. Thus, the loss becomes
> \\[\log\left( \sum_v \exp(e_v^\top W_{ov}x_1) \right) - \log\exp(e_i^\top W_{ov} x_1) = -\log\frac{ \exp(e_i^\top W_{ov} x_1) }{ \sum_v \exp(e_v^\top W_{ov}x_1) }.\\]
> Thank you for your question and we will fix the typo in the revision.
>
> **Q2:** In Section 6, the authors theoretically prove the generalization ability of the trained Transformer. Can the generalization ability be proved in the experimental part?
>
> **A2:** Yes. Please see the attached PDF file in the global response for the test loss result. For the test dataset, we randomly construct test data as described in Theorem 3, and plot the average test loss for 20 trials. As shown in the figure, the test loss also converges almost to 0. In addition, since the generalization ability is determined by the parameter $W_{kq}$ and $W_{ov}$,  alignment of these parameters with their corresponding max-margin solutions as shown in Figure 2 of the paper implies the generalization ability as well.
>
> **Q3:** Why do the authors design a two-stage training algorithm? Is it valuable in practical applications compared to single-stage training?
>
> **A3:** Thanks for the question. Our characterization of the realizable dataset is via two steps: (i) identifying collocation and (ii) using collocation to define partial ordering and optimal token. This naturally motivates the two training stages to learn the corresponding information: (i) stage 1 learns collocation, i.e., next-token prediction for length-one sentences; and (ii) stage 2 learns partial ordering to identify optimal token. Thus, it is also meaningful to apply such a two-stage algorithm in practice. We also expect that stage 1 of pre-processing value matrix using one-token sentences could help to stablize the entire training process of transformers. Single-stage training may still yield good solution simply from optimization perspective, and we are currently working on analytically characterizing the dynamics of such a training process.
>
> **Q4:** This manuscript simplifies the Transformer into a single layer for analysis. While I understand the need for this simplification, it would be nice to include a brief paragraph outlining the views on generalizing to multi-layer Transformers.
>
> **A4:** Thanks for the suggestion. A possible approach to generalize it into multi-layer transformers is to view our single layer as the last layer of a multi-layer transformer. Therefore, the input $X$ in the paper can be viewed as the output of a multi-layer network. Then, we can combine our technique for the last layer and the technique of neural tangent kernel (NTK) for other layers (Emami et al., 2021), and analyze the training dynamics of such a multi-layer transformer. We expect that the last layer evolves similarly to our findings given that other layers output $X$ satisfies certain conditions, which may be potentially shown by exploiting the realizability of the dataset and the properties of NTK.
>
> Emami et al. "Implicit bias of linear RNNs." ICML 2021.
>
> ---
>
> We thank the reviewer again for the highly inspiring comments. We hope that our responses have resolved your concerns. If so, we wonder if the reviewer could kindly consider to increase your score. Certainly, we are more than happy to answer your further questions.

---

> > ### Comment · Reviewer_jsSk · 2024-08-11
> >
> > Thanks to the authors for replying to my questions. I have read the relevant comments and will maintain this score.

---

> > > ### Author Response · Authors · 2024-08-12
> > > **Thank you**
> > >
> > > Many thanks for your continual and strong support of our paper!

---

### Official Review · Reviewer_ZtZ3 · 2024-07-11

**Soundness:** 3
**Presentation:** 3
**Contribution:** 2
**Rating:** 6
**Confidence:** 3

**Summary:**

In this work, the authors mathematically examine the learning dynamics of simple transformers for next token prediction. To allow a mathematical analysis, they consider a  highly simplified setting: the transformer consists of a single attention layer followed by a single feed-forward layer, the layers are trained one after the other (i.e., in a custom, decoupled training procedure), and the dataset adheres to a set of mathematical properties that make sure the loss can be arbitrarily close to zero. They show that both layers converge in direction to their corresponding max-margin solutions sub-linearly.

**Strengths:**

The paper adresses an important problem, namely how we could/should understand the learning dynamics of transformers. The paper seems a valid contribution to this problem, and provides a step in the direction of a better theoretical understanding.

The paper defines a set of mathematical properties a dataset for next-token-prediction should adhere to, to obtain a training error arbitrarily close to zero. This might be useful for future work in the community.

The paper is well-written, well-structured, the related works section is extensive and the work is put into the context of previous works.

Great care has been taken in the mathematical formulations and proofs; I wasn’t able to check the details, but the mathematical derivations seem correct and they are extensive.

**Weaknesses:**

The paper completely lacks a discussion of how general the obtained insights are; i.e., what the paper contributes beyond the setting of the simple network architecture, custom dataset and custom training method used to derive the results. It is understandable that assumptions and simplifications have to be made to make a problem amenable to a mathematical analysis, but the paper contains no discussion of these limitations nor any experiments to further examine this. This also makes it hard to judge the actual contribution of the work.

The current version of the paper is only easy to read for people working on this exact topic. To also be of interest to a somewhat broader audience (e.g., machine learning experts with a good grasp of mathematics), some concepts would need to be better motivated and/or explained in a more intuitive way. E.g., although the text mentions, ‘We first provide some intuitions about those two properties’ (L145), there seems to be no real intuitive explanation of why collocations and partial orders are introduced, and how this should be interpreted. The definition and arguments are purely mathematical. The example dataset provides some additional insights, but does not clarify why these properties are important, how this leads to zero training loss in the limit, etc…

**Questions:**

How would you change the manuscript to make it more easy to understand your definitions and motivation behind the mathematics?

How would you change the manuscript to better assess/discuss the limitations of your work?

**Limitations:**

See above, no negative societal impact.

---

> ### Author Rebuttal · Authors · 2024-08-07
>
> We thank the reviewer for the helpful comments, and would like to provide the following responses.
>
> **Q1:** How would you change the manuscript to make it more easy to understand your definitions and motivation behind the mathematics? For example, the example dataset provides some additional insights, but does not clarify why these properties are important, how this leads to zero training loss in the limit, etc.
>
> **A1:** Thanks for the suggestion. We will provide the following high-level description of our dataset and their properties.
>
> The goal of introducing the collocation and query-dependent partial order is to make training transformer provable and interpretable. We breakdown the insights as follows.
>
> (i) Since the model should apply to all subsequences of sentences in the training set including length-two sentences, then collocation between two words naturally becomes a property of the dataset.
>
> (ii) It can be observed that the input of our loss function is a convex combination of $L$ points each being a projection of one token embedding in the sentence, where the weights in the convex combination are the attention scores of each token with the query token. In this way, the loss values of these tokens determine an order among them, and the token that achieves the minimum loss value serves as the optimal token. In other words, some tokens has larger attention scores than others under a specific query. These facts motivate us to introduce query-dependent partial order.
>
> Combining (ii) with (i), the collocated token of the optimal token becomes the predicted next token. Finally, due to the injection property of collocation and the structure of cross-entropy, the loss function can be trained to approach zero.
>
> **Q2:** How would you change the manuscript to better assess/discuss the limitations of your work? For example, provide discussion of how general the obtained insights are; i.e., what the paper contributes beyond the setting of the simple network architecture, custom dataset and custom training method used to derive the results.
>
> **A2:** Thanks for the suggestion. We provide the following discussion on the generality of our findings and will add it to our revision.
>
> We remark that our work establishes the basic framework to analyze the convergence performance of transformers for NTP task. Several insights can be extended to more complex settings.
>
> Regarding multi-layer transformers, for example, one possible idea is to view the single layer in our work as the last layer of a multi-layer transformer. This is promising if we consider the last layer training regime (Kirichenko, et al., 2022). In this regime, if the output of other layers satisfies certain conditions such as near-orthogonality, which is often satisfied in high dimensional setting, our convergence and generalization results can be directly applied to multi-layer transformers.
>
> To generalize dataset, for example, our technique can be extended to study a noisy version of collocation. We expect that most of our proof steps can still be applied by incorporating appropriate noise bound.
>
> To generalize training method, an alternative natural method is to update the linear layer and the attention layer simultaneously instead of two-stage training. A popular way to implement such an algorithm is via two timescale stepsizes so that the attention update can be in a faster scale and the final convergence can be determined by the feedforward layer by incorporating the suboptimality error of the attention layer. Such suboptimality error can be bounded by our current techniques for training stage 2.
>
> Kirichenko et al. "Last layer re-training is sufficient for robustness to spurious correlations." arXiv:2204.02937, 2022.
>
> ---
>
> We thank the reviewer again for the highly inspiring comments. We hope that our responses have resolved your concerns. If so, we wonder if the reviewer could kindly consider to increase your score. Certainly, we are more than happy to answer your further questions.

---

> > ### Author Response · Authors · 2024-08-10
> > **A gentle reminder**
> >
> > Dear Reviewer ZtZ3
> >
> > We've taken your initial feedback into careful consideration in our response. Could you please check whether our responses have properly addressed your concerns? If so, could you please kindly consider increasing your initial score accordingly? Certainly, we are more than happy to answer your further questions.
> >
> > Thank you for your time and effort in reviewing our work!
> >
> > Best Regards, Authors

---

> > > ### Comment · Reviewer_ZtZ3 · 2024-08-12
> > >
> > > I have carefully read the rebuttal and, especially following the effort to suggest approaches to generalize the result, I will raise my score from 5 to 6.

---

> > > > ### Author Response · Authors · 2024-08-12
> > > > **Thank you**
> > > >
> > > > Many thanks for your increase of the score and strongly support of our paper!

---

### Official Review · Reviewer_72WK · 2024-07-16

**Soundness:** 3
**Presentation:** 3
**Contribution:** 3
**Rating:** 6
**Confidence:** 4

**Summary:**

The paper presents a non-asymptotic analysis of training dynamics for a single-layer transformer used in next-token prediction tasks. It introduces a two-stage training algorithm leveraging structural properties of the training dataset, defined via collocations and query-dependent partial orders. The findings include sub-linear convergence of both the feed-forward and self-attention layers to their respective max-margin solutions, and a linear convergence rate for the cross-entropy loss, supporting non-trivial prediction capabilities on unseen data. The approach is validated through theoretical analysis and empirical results, enhancing understanding of transformers' training and generalization behaviors.

**Strengths:**

1. The paper introduces novel theoretical frameworks for analyzing transformer training, focusing on non-asymptotic convergence.
2. The research is technically robust, with sound mathematical derivations and empirical validation.
3. The concepts and results are communicated effectively, albeit with room for improvement in some technical descriptions.

**Weaknesses:**

1. Some of the mathematical concepts, particularly the query-dependent partial orders, are complex and could be better explained.
2. While the theoretical results are strong, additional empirical studies, particularly on real-world datasets, could further strengthen the claims.

**Questions:**

What are the potential implications of the findings on transformer training efficiency and computational costs in practical settings?

**Limitations:**

NaN

---

> ### Author Rebuttal · Authors · 2024-08-07
>
> We thank the reviewer for the helpful comments, and would like to provide the following responses.
>
>
> **Q1:** Some of the mathematical concepts, particularly the query-dependent partial orders, are complex and could be better explained.
>
> **A1:** We appreciate the feedback on the complexity of the mathematical concepts. To address this, we will provide more intuitive explanations and examples in the revised manuscript.
>
> Specifically, query-dependent partial order formalizes the concept of importance or relevance of tokens in a given sentence. If one token is more important than others or more relevant to the next token, it is considered as an optimal token and thus is ''greater''  than other tokens.  For example,  assume that the sentence *Machine learning is a popular* has next word *area*, and *popular* is the most important word. Then, under the query *popular*, *popular* itself is the optimal token, and hence it is ''greater''   than other tokens.
>
>
> In addition, such a partial order is **query-dependent** so that the order can be different given different queries. For example, consider a new sentence *A popular area is machine* with next word *learning*. Then, under the query *machine*, *machine* itself is more important or ''greater'' than *popular*.
>
> Finally, the motivation for defining these concepts is detailed in Response A1 to Reviewer ZtZ3.
>
> **Q2:** While the theoretical results are strong, additional empirical studies, particularly on real-world datasets, could further strengthen the claims.
>
> **A2:** Thanks for the suggestion. We are actively seeking suitable real-world datasets and experiments to further illustrate our results.
>
> **Q3:** What are the potential implications of the findings on transformer training efficiency and computational costs in practical settings?
>
> **A3:** Thanks for the question. Our results have the following implications. (1) The collocation training can be separated from the attention training to reduce the complexity of joint training, and at the same time achieve stable and fast convergence rate for each separate training. (2) Our result shows that the loss function converges fast (with potentially linear convergence rate), and hence the training does not require many iterations. The main computational costs lie in the gradient computations in each iteration, which scales with the number of training parameters. Thus, employing techniques such as LoRA to reduce the number of training parameters is crucial to achieve scalable training.
>
> ---
>
> We thank the reviewer again for the highly inspiring comments. We hope that our responses have resolved your concerns. If so, we wonder if the reviewer could kindly consider to increase your score. Certainly, we are more than happy to answer your further questions.

---

> > ### Comment · Reviewer_72WK · 2024-08-10
> >
> > I have thoroughly reviewed all the comments and the author’s responses, and I will remain positive about this submission.

---

> > > ### Author Response · Authors · 2024-08-12
> > > **Thank you**
> > >
> > > Many thanks for your continual and strong support of our paper!

---

### Official Review · Reviewer_U8Jg · 2024-07-18

**Soundness:** 2
**Presentation:** 3
**Contribution:** 2
**Rating:** 5
**Confidence:** 3

**Summary:**

This paper conducts a non-asymptotic analysis of training dynamics for a one-layer transformer, focusing on next-token prediction tasks.
It provides a mathematical framework  based on partial order to formally characterize a realizable training dataset for next-token prediction.
It also introduces a two-stage training algorithm that ensures fast convergence, with both layers approaching their max-margin solutions sub-linearly.

**Strengths:**

The paper develops a detailed theoretical framework that both analyzes  non-asymptotic convergence of the training and generalization capabilities of transformers in the next-token prediction task.

**Weaknesses:**

The approach described in the paper may not align with practical Transformer training.
Please see the questions below for more details.

**Questions:**

Q1: Although the ground truth model is deterministic, it seems uncommon in machine learning to assume a statistical model because the next word is not deterministically determined from a deterministic context. Besides assuming n, where is the existence of p*L theoretically necessary?
Q2: The assumption of the existence of collocations seems too far removed from reality. What are some realistic data where such assumptions hold? The setting in Dryer, 1991 seems artificial. Especially, the existence of n() is crucial for the two-stage training and the proof, and it forms a major assumption at the core of this paper. Is it possible to actually prepare n() in real datasets?
Q3: I am confused about the definition of partial order. In Definition 1, it is stated that if there is at least one sentence where x >{x_q} x' and n(x) = L_{L+1} != n(x'), then it is also permissible for there to be sentences where n(x) != L_{L+1} = n(x'). I could understand it if it were defined to hold for all sentences.
Q4: How commonly is the normalized gradient used in actual Transformer training? In the training of Transfomer, AdamW is typically used.
Q5: Assumption 3 may not hold in situations where attention is sparse. It is important that attention is placed on the optimal token, but it is known that the attention map in actual Transformers is generally sparse. Doesn’t this assumption contradict the sparsity of attention?

**Limitations:**

The assumption of the existence of collocations seems far from real traninng dataset.

---

> ### Author Rebuttal · Authors · 2024-08-07
>
> We thank the reviewer for the helpful comments.
>
> **Q1:** Deterministic ground truth model seems uncommon. Besides assuming n, where is the existence of $p_L^*$ theoretically necessary?
>
> **A1:** In recent line of theoretical research on transformers, deterministic models are often adopted such as in Li et al., (2024) on next-token prediction and Tarzanagh et al., (2023) on binary classification, mainly because deterministic models can offer
> valuable theoretical insights due to their simplicity and clean structure. In particular, it provides a clear cause-and-effect relationship between inputs and outputs, which makes it appealing to facilitate the interpretation of the model. Note that deterministic model of ``separable data'' is also commonly studied in theoretical works on neural networks such as in Soudry et al., (2018).
>
> The ground truth model $p^*_L$ serves as a theoretical baseline for analyzing an algorithm so that the performance of a training algorithm is well defined. As a notation, $p^*_L$ can be replaced by $\mathrm{n}(X)$, where $X$ contains $L$ tokens. As a realizable model, it is captured by collocation and partial order. With the existence of such a model $p^*_L$, the training algorithm can have provable performance.
>
> Tarzanagh et al. Transformers as support vector machines. arXiv:2308.16898, 2023.
>
> Li et al. Mechanics of next token prediction with self-attention. AISTATS 2024.
>
> Soudry et al. "The implicit bias of gradient descent on separable data." JMLR 2018.
>
> **Q2:** Existence of collocations seems far from reality; realistic data where such assumptions hold? How to prepare n() in real datasets?
>
> **A2:** Thanks for the question. The existence of collocations can be observed in many natural languages, where certain words frequently appear together (Lehecka 2015). Taking such an assumption enables us to derive interesting theoretical results, while remaining arguably reasonable in practice. Note that the assumption of separable data, as a special case of this, has formed an important line of theoretical research in deep learning to explore the implicit bias and training dynamics of neural networks (Soudry et al. 2018, Chizat and Francis 2020).
>
> Given a dataset, $\mathrm{n}$ can be prepared by the counting method described in Lehecka, T. (2015). More generally, it can be extracted by simply including all two-token sequences from the dataset, and each second token $y$ is the next token of the first token $x$, i.e., $n(x)=y$. Furthermore, leveraging domain-specific knowledge, such as medical dictionaries, can also enrich collocations. These approaches allow for the practical extraction of \( \mathrm{n}() \) in a scalable and efficient manner.
>
> Lehecka. Collocation and colligation. In Handbook of pragmatics online. Benjamins 2015.
>
> Chizat and Francis Implicit bias of gradient descent for wide two-layer neural networks trained with the logistic loss. COLT 2020.
>
> **Q3:** How to understand definition of partial order.
>
> **A3:** Thanks for the question. We clarify that the partial order is **query-dependent**. Therefore, if $x>\_{x_q} x'$ and $n(x)=x_{L+1}\neq n(x')$ in some sentences ending with query $x_q$, it is **not** possible that $n(x)\neq x_{L+1} = n(x')$ in another sentence ending with the same query $x_q$.
>
> This is because of the structure of the transformer where the attention weights are proportional to $\exp(x^\top W_{kq} x_q)$. If $n(x)=x_{L+1}\neq n(x')$ under query $x_q$,  then $x^\top W_{kq} x_q > (x')^\top W_{kq}x_q$ as long as the sentence ends with $x_q$. Thus, $x$ is always more important than $x'$ in $x_q$-ended sentences. However, $n(x)\neq x_{L+1} = n(x')$ **is** still possible in other sentences ending with **different queries** than $x_q$.  These facts motivates us to introduce *query-dependent* partial orders.
>
> **Q4:** How commonly is the normalized gradient used? AdamW is typically used.
>
> **A4:** Thanks for the question. Although normalized gradient descent (NGD) is often favored in theoretical studies, there are also empirical works showing that the achieved accuracy of NGD is slightly better than Adam in training transformers (Cutkosky and Mehta 2020), where momentum is employed to improve the performance of NGD. Cutkosky and Mehta (2020) also theoretically proved that NGD achieves a fast convergence rate for generic optimization, which aligns with our theoretical findings that NGD achieves a linear convergence rate for NTP with one-layer transformers.
>
> We acknowledge that AdamW is indeed more commonly used in practice. However, it is much more challenging to analyze the training dynamics under AdamW theoretically. So far, all theoretical analysis on transformers have been focused on GD and its variants. It is somewhat necessary to first develop techniques for GD and its variants, before these tools can be further advanced to study AdamW. Our work takes a first step to understanding the NGD dynamics of Transformers for NTP, and we are actively exploring the analysis for Adam-based Transformer training.
>
> Cutkosky and Mehta, Momentum Improves Normalized SGD, ICML 2020.
>
> **Q5:** Assumption 3 may not hold in situations where attention is sparse.
>
> **A5:** Thanks for the question. We apologize that the statement of Assumption 3 may cause confusion. It actually requires that the number of optimal tokens is greater than or equal to the number of **each individual** non-optimal tokens, not the summation of all non-optimal tokens. Hence, such an assumption is not in conflict with sparse attention. One simple example satisfying Assumption 3 is a sentence where every token is distinct. The transformer's attention needs to focus only on the single optimal token, which is clearly sparse.
>
> ---
> We thank the reviewer again for the highly inspiring comments. We hope that our responses resolved your concerns. If so, we wonder if the reviewer could kindly consider to increase your score. Certainly, we are more than happy to answer your further questions.

---

> > ### Author Response · Authors · 2024-08-10
> > **A gentle reminder**
> >
> > Dear Reviewer U8Jg,
> >
> > We've taken your initial feedback into careful consideration in our response. Could you please check whether our responses have properly addressed your concerns? If so, could you please kindly consider increasing your initial score accordingly? Certainly, we are more than happy to answer your further questions.
> >
> > Thank you for your time and effort in reviewing our work!
> >
> > Best Regards,
> > Authors

---

> > > ### Author Response · Authors · 2024-08-12
> > > **A gentle reminder before the discussion period ends**
> > >
> > > Dear Reviewer U8Jg,
> > >
> > > As the author-reviewer discussion period will end soon, we would like to check whether our responses have properly addressed your concerns? If so, could you please kindly consider increasing your initial score accordingly? Certainly, we are more than happy to answer your further questions.
> > >
> > > Thank you for your time and effort in reviewing our work!
> > >
> > > Best Regards,
> > > Authors

---

> > ### Comment · Reviewer_U8Jg · 2024-08-13
> >
> > Let me confirm about "collocations".
> > Please tell me how to construct n in real data.
> > It seems like it needs to be defined for every word, not some words and also, n seems to be a deterministic function.
> > Considering the above, it seems that there exists an n such that every word can definitively find one next token.

---

> > > ### Author Response · Authors · 2024-08-13
> > >
> > > Thank you for your insightful questions. Below we answer the reviewer's questions.
> > >
> > > **Can every word find a next token?** We do not need every word to find a next token. If a token $x$ does not have next token, then $x$ falls into the category of non-optimal tokens or non-comparable tokens (see lines 182-184 of the paper). Then the trained transformer will not attend to such a token. Thus, $x$ won't play a role to predict next tokens in sentences. However, it is still possible that other tokens can predict $x$ as a next token.
> > >
> > > **Deterministic function:** Yes, n is assumed to be deterministic. One reason is that in real world settings, n can be prepared to be a deterministic mapping (see below about construction). In recent line of theoretical research on next-token prediction, deterministic function is commonly adopted, such as in (Li et al., 2024, Tarzanagh et al., 2023.), as an informative model to enable tractable analysis for transformers. Such a model is also analogous to the deterministic structure of 'separable data' widely taken to develop deep learning theory in the literature such as in (Soudry et al, 2018, Taheri et al., 2023).
> > >
> > > **Construct n in real data:** We can construct the collocation n by employing various standard techniques developed in linguistic analysis (Lehecka, 2015). To elaborate a simple version (those practical techniques can include more sophisticated tricks), by processing all sentences in the corpus as detailed in (Lehecka, 2015), the frequency of each 'ordered' word pair $(x,y)$ that appears in the same sentence can be calculated. Then, the most frequent $y$ paired with each $x$ is chosen as its collocated word, i.e., let $n(x)=y$. Such method includes our original construction from length-2 sentences as a special case. There are also several off-the-shelf NLP tools to construct collocations such as Natural Language Toolkit (NLTK).
> > >
> > > We hope that our responses have resolved your concerns. If so, we kindly ask the reviewer to consider increasing the score accordingly. If possible, we also kindly ask the reviewer to evaluate the paper based on the theoretical contributions and the novel mathematical techniques that we develop to analyze the next-token prediction problems, which can be applied to studying more sophisticated models in the future. Certainly, we are more than happy to answer your further questions.
> > >
> > >
> > > Lehecka. Collocation and colligation. In Handbook of pragmatics online. Benjamins 2015.
> > >
> > > Li et al. "Mechanics of next token prediction with self-attention." AISTATS 2024.
> > >
> > > Soudry et al. "The implicit bias of gradient descent on separable data." JMLR 2018.
> > >
> > > Taheri et al. "On generalization of decentralized learning with separable data." AISTATS 2023.
> > >
> > > Tarzanagh et al. "Transformers as support vector machines." arXiv:2308.16898, 2023.

---

> > > > ### Comment · Reviewer_U8Jg · 2024-08-14
> > > >
> > > > Thank you for clarification. I raised my score.
> > > > However, I am not satisfied with the composition of n. If we use the frequency of two words, we should consider the probability distribution based on the frequency rather than the maximum one. It seems important as a language model that a word expresses multiple meanings by co-occurring with other words.

---

### Author Rebuttal · Authors · 2024-08-07

We thank all reviewers for their feedback, which will greatly improve our paper.

In the attached PDF file, we provide a figure for an additional experiment verifying the generalization ability described in Theorem 3, as a response to the second question Q2 of Reviewer jsSk. In the experiment, we construct a test dataset described in Theorem 3 and plot the average test loss of 20 times against the training loss. As shown in the figure in the attached PDF, the test loss also converges almost to 0.

---

### Decision · Program_Chairs · 2024-09-25

**Decision:**

Accept (poster)

**Comment:**

In this submission, the authors present interesting theoretical insights into transformer training dynamics for next-token prediction, supported by detailed mathematical analysis. Although the setting is simplified, the reviewers concur that it contributes to understanding transformers. The authors have also addressed reviewers' concerns  in their rebuttal. While further empirical validation on real-world datasets would enhance the work's impact, the theoretical contributions in their current form provide a solid foundation for advancing the understanding of transformer training dynamics. To sum up, the decision is to accept the paper.